# Multitemporal analysis of Sentinel-1 backscatter during snow melt using high-resolution field measurements and radiative transfer modelling

Francesca Carletti<sup>1,2</sup>, Carlo Marin<sup>3</sup>, Chiara Ghielmini<sup>1</sup>, Mathias Bavay<sup>1</sup>, and Michael Lehning<sup>1,2</sup>

**Correspondence:** Francesca Carletti (francesca.carletti@slf.ch)

Abstract. The spatiotemporal evolution of snow melt is fundamental for water resources management and risk mitigation in mountain catchments. Synthetic Aperture Radar (SAR) images acquired by satellite systems such as Sentinel-1 (S1) are promising for monitoring wet snow due to their high sensitivity to liquid water content (LWC) and ability to provide spatially distributed data at a high temporal resolution. While recent studies have linked multitemporal S1 backscattering to snow melt phases, a correlation with detailed snowpack properties is still missing. To address this, we collected the first dataset of comprehensive wet snow properties tailored for SAR applications over two consecutive snow seasons at the Weissfluhjoch field site near Davos, Switzerland. First, we tested previous methods which use multitemporal S1 backscattering to characterize melting phases, and demonstrated that the observed monotonous increase in backscattering following the local minimum is due to the development of surface roughness. Then, we used the measured snow properties as input to the Snow Microwave Radiative Transfer (SMRT) model to reproduce S1 backscattering signals. Our simulations showed that rather than melting phases, time series of backscattering rather identify regimes dominated by either LWC, early in the season, or surface roughness, later on. The results also highlight several key challenges for reconciling S1 signals with radiative transfer simulations of wet snow: (i) the discrepancy in spatiotemporal variability of LWC as seen by the satellite and validation measurements, (ii) the lack of fully validated permittivity, microstructure and roughness models for wet snow in the C-band, (iii) the difficulty of capturing wet snow features potentially generating stronger scattering effects on a large scale – such as internal snowpack structures, soil features in case of low LWC, and surface roughness – which are not necessarily captured by point-wise measurements.

#### 1 Introduction

Seasonal snowpack in mountain catchments is one of the most important water resources, as it accumulates and stores water during winter and releases it consistently in the form of runoff during the melting period (Viviroli and Weingartner, 2004). In alpine streams, discharge is largely dominated by snow melt from May to July and more than one sixth of the world's population relies on meltwater released from higher altitudes for drinking water, crop irrigation and hydropower production (Beniston et al., 2018). However, melting snow can also cause wet- and glide-snow avalanches (Bellaire et al., 2017; Fromm

<sup>&</sup>lt;sup>1</sup>WSL Institute for Snow and Avalanche Research SLF, Davos, Switzerland

<sup>&</sup>lt;sup>2</sup>School of Architecture, Civil and Environmental Engineering, École Polytechnique Fédérale de Lausanne EPFL, Lausanne, Switzerland

<sup>&</sup>lt;sup>3</sup>Institute for Earth Observation, Eurac Research, Bolzano, Italy

et al., 2018), which pose significant threats to human life and infrastructures. Additionally, rain-on-snow events on already wet snowpacks are linked to increased runoff and shorter time lags between precipitation onset and the resulting runoff (Würzer et al., 2016). These events can have catastrophic consequences and their occurrence is supposed to increase in response to a sustained warming (Beniston and Stoffel, 2016). Therefore, information about the spatiotemporal evolution of snow melt is beneficial for both the management of water resources and for risk mitigation.

Identifying wet snow is complex both when using manual measurements, automatic instruments and physics-based snow models. Datasets of manual measurements of snow water equivalent (SWE) and liquid water content (LWC hereafter) at high temporal resolution are generally rare due to the time, effort and resources required for their collection. There have been considerable advances in technologies that use the dielectric properties of snow in the microwave range to estimate LWC in a non-destructive way (Schmid et al., 2014; Koch et al., 2014). However, the application of these methods is limited to one single point without the possibility to capture the spatial variability of the processes. Additionally, their installation and maintenance is often complicated and expensive, and the extraction of the physical parameters is usually hindered by noise. physics-based layered snow models like the SNOWPACK-Alpine3D model chain (Bartelt and Lehning, 2002; Lehning et al., 2006) or GEOtop (Endrizzi et al., 2014) are used to overcome these challenges, as they can simulate LWC and SWE at high spatial and temporal resolutions based on meteorological forcings. However, meteorological forcings also represent a major source of uncertainty – especially when needed at high spatial resolution – affecting the accuracy of the results (Raleigh et al., 2015). This adds up to the uncertainties related to the amount and type of parametrizations used (Günther et al., 2019).

40

In this context, a valuable opportunity to identify wet snow is offered by synthetic aperture radar (SAR hereafter) systems. SAR measurements are highly sensitive to the free liquid water contained in wet snow (Nagler and Rott, 2000). At certain frequencies, the increase in liquid water generates high dielectric losses and increased absorption coefficients (Denoth et al., 1984; Sihvola and Tiuri, 1986; Mätzler, 1987; Ulaby et al., 2014). Therefore, the radar backscatter drops to lower intensities with respect to winter averages (Ulaby et al., 1987; Strozzi et al., 1997; Strozzi and Matzler, 1998; Nagler and Rott, 2000; Ulaby et al., 2014; Nagler et al., 2016; Lin et al., 2016). This raised the question of whether different types of snow cover could be classified based on their response to active microwave signals. This challenge has been addressed with various approaches over the years. Between 1993 and 1995, at the field site of Weissfluhjoch in the Swiss Alps, Strozzi et al. (1997); Strozzi and Matzler (1998) conducted tower-based C-band radiometric measurements at all polarizations across a wide range of incidence angles. Simultaneously, they carried out monthly measurements of snow physical properties. These measurements were used to classify the observed snow covers into categories ranging from dry snowpacks, to thin moist layers overlying dry snow, to wet snowpacks with either smooth or rough surfaces. Relying on a tower-based radiometer, the experiments were highly controlled, allowing detailed investigation of radar responses to each snow condition. Nevertheless, significant sources of uncertainty remained – especially the influence of surface roughness on wet snow surfaces, which was not quantitatively measured, but only qualitatively assessed. These detailed studies, along with the work of Kendra et al. (1998), raised questions about theoretical foundations and systematic reliability of LWC retrieval algorithms based on C-band full-polarimetric SAR imagery, which had been developed shortly before (Shi et al., 1993; Shi and Dozier, 1995). In particular, the scattering mechanisms assumed in these retrievals may have been biased by a combination of conditions that strongly favored surface scattering. Extending the prior knowledge to a spatial and multitemporal context, (Nagler and Rott, 2000) developed an algorithm based on repeat-pass SAR images to map wet-snow in mountainous areas, defining a backscatter drop of 3 dB to distinguish wet snow from other surfaces. Comparisons with snow maps from different sources showed generally good agreement above the snow line, but consistent biases in areas with fragmented snow cover.

After a progress freeze due to the scarcity of SAR data in the past and simultaneous field measurements, the research interest in the topic was renewed since the launch of the Sentinel-1 (S1 hereafter) joint mission of the European Space Agency (ESA) and the European Commission in 2014. At alpine latitudes, S1 acquires C-band SAR imagery in the early morning and late afternoon, regardless of the weather, with a revisit time of 6 days. The SAR imagery is available free of charge. Marin et al. (2020) used these images for the first time to develop a correlation between the multitemporal S1 SAR backscatter and the snow melt dynamics. Over 5 different alpine sites, the authors have found that the multitemporal S1 SAR acquisitions allow the detection of the melting phases, i.e. moistening, ripening and runoff (Dingman, 2015) with a good agreement with insitu observations and layered, physics-based snow models. In particular, the backscatter decreased as soon as liquid water appeared in the snowpack and increased progressively and simultaneously with the runoff release. Deriving and applying a set of identification rules, the authors could define the melting phases for the test sites with relatively small lag errors with respect to the revisit time of S1. Consequently, local minima in S1 multitemporal backscatter time series and sharp increases thereafter were associated with snowpack saturation, the onset of runoff, and snow ablation (Darychuk et al., 2023; Gagliano et al., 2023).

These approaches hold great potential for monitoring the temporal evolution of the melting dynamics, particularly over wide and scarcely instrumented areas. However, to fully use the multitemporal information provided by S1 for snow melt monitoring, a deeper understanding of the underlying scattering mechanisms – especially the role of surface roughness (Marin et al., 2020) – is still required. Specifically, knowing the time window in which different scattering effects dominate and under which conditions the C-band radar backscatter is fully absorbed by the melting snowpack would enable to extract as much information as possible from S1 time series. To date, the only effort in this direction was made by Brangers et al. (2024) using tower-based C-band measurements. However, this study lacks high-temporal-resolution ground-truth validation with measured snow properties. Moreover, comparisons with S1 were hindered by several factors, including sensor calibration issues and the small footprint size – which likely introduced speckle noise and failed capturing larger-scale scattering processes.

75

Overall, the main limitation to improving the understanding of the interaction of S1 backscatter signals with melting snow cover is the lack of reference ground data. Over alpine snowpacks, it is common to observe the formation of ice layers either at the surface (Quéno et al., 2018) or at deeper snowpack depths (Pfeffer and Humphrey, 1998). Moreover, in temperate alpine areas characterized by high snow accumulation and intense solar radiation, suncups may form spontaneously on the snow surface during the ablation season (Post and LaChapelle, 2000; Mitchell and Tiedje, 2010), increasing the surface roughness significantly (Fassnacht et al., 2009). These phenomena are known to impact the radar response to wet snow (Shi and Dozier, 1995; Strozzi and Matzler, 1998; Kendra et al., 1998; Nagler and Rott, 2000; Yueh et al., 2009).

However, high-resolution and detailed snow measurements alone are insufficient to address this issue. It is equally important to rely on a method to interpret them from a radar perspective. A promising and increasingly adopted approach involves the

use of state-of-the-art radiative transfer (RT hereafter) models. Picard et al. (2018) developed the Snow Microwave Radiative Transfer (SMRT) model, a versatile model that can be used in active and passive mode to compute backscatter and brightness temperature from multilayered media such as snowpacks or ice sheets overlying reflective surfaces, e.g. ground, ice, or water. SMRT responds to the need of a modular and flexible approach to unify and compare the wide range of pre-existing representations of microstructure, electromagnetic theories, soil models and permittivity formulations. While wet snow holds significant importance for various applications, both SMRT and other similar models were primarily developed and validated for dry snow conditions in Arctic and Antarctic snowpacks, or ice sheets (Proksch et al., 2015; Rott et al., 2021; Soriot et al., 2022; Meloche et al., 2022; Husman et al., 2023). Both the vertical structure and the surface of these types of snowpack are often less complex than that of a seasonal alpine snowpack. To date, the above mentioned ensemble of complex melting snowpack processes has been scarcely investigated by means of radiative transfer models due to the lack of ground reference data (Shi and Dozier, 1995; Strozzi et al., 1997; Kendra et al., 1998; Nagler and Rott, 2000; Magagi and Bernier, 2003; Lodigiani et al., 2025). Murfitt et al. (2024) recently used SMRT to explore, for the first time, the temporal evolution of the interaction between wet snow and radar waves in a study on lake ice melt. However, the radiative transfer modelling of wet snow still lacks dedicated effort and validation.



The objective of this work is to collect the first ground reference dataset on melting snow tailored for SAR applications and to use it together with SMRT to better understand the key processes governing the backscatter signatures recorded by S1. Previously, only Lund et al. (2022) carried out a similarly extensive snow pit campaign in coordination with S1 passages. While this study helped advance the interpretation of S1 backscatter responses to diurnal snowpack variations, important scattering properties such as the optical diameter and the surface roughness were not measured. As a result, interpreting these measurements from the radar perspective – and consequently comparing them with S1 acquisitions – was not possible. In our work, we focus on the co-polarized vertical backscattering only, due to its high signal to noise ratio for wet snow (Naderpour et al., 2022) and to the fact that, due to the partial implementation of some of the key processes, it is not possible to simulate accurate cross-polarized backscattering responses with the current version of SMRT. To our knowledge, this is the first attempt to translate ground measurements – specifically designed for RT modelling, including wetness and roughness - into radar signals using SMRT to reproduce and interpret S1 acquisitions over a wet, multilayered alpine snowpack. This research provides valuable insights in two main areas. First, it advances the understanding of the interaction between S1 radar backscatter and wet snow. Specifically, it reveals the effects of spatiotemporal variability of LWC within the S1 footprint occurring between satellite and measurement acquisitions. It also describes the impact of surface roughness on backscatter signatures and highlights challenges in capturing key wet snow conditions that likely generate scattering at wider-scales. These include internal snowpack structures, large-scale surface roughness, and interactions with the wet soil interface when the snowpack is only slightly wet. Second, the study addresses the RT modelling of melting, layered snowpacks, highlighting the current lack of fully validated permittivity and roughness models for wet snow at C-band frequencies. With ground reference data and adequate process understanding and modelling, RT models like SMRT may evolve in tools to interpret and translate the information contained in multitemporal SAR backscatter into valuable input for snow-hydrological modelling.

**Figure 1.** Location of the Weissfluhjoch field site with respect to Swiss national borders (a) and the town of Davos (b). The designated area for snow profiles is shown in (c) under semi-snow-free conditions (picture taken in Sep 2024, camera oriented towards the north-east), enclosed by a flagged fence. It is worth noting that only a portion of this fenced area was effectively used for snow profiles. Picture (d) shows the typical snowpit measurement setup.

#### 2 Campaign overview



This work builds upon a dataset of 85 snow pits collected during a two-season campaign (2022-2023 and 2023-2024) at the high-altitude Weissfluhjoch Versuchsfeld (WFJ) field site, located in the Rhaetian Alps near Davos, Switzerland. The measurement field lies at an altitude of 2536 m a.s.l. on a relatively flat area embedded in a south-east facing valley. The site is partially wind sheltered from a small hill situated on the south-east – however, the dominant wind blows from northwest, in addition to katabatic wind. For this measurement campaign, we secured a protected field covering approximately two times the footprint area of S1, i.e.  $20 \times 20$  m. However, only a portion of this field was effectively used for measurements, while the remaining area was consistently left undisturbed. The secured field has a light slope value between 2 and 7%. The flatness of the terrain is fundamental for the study of the interaction between wet snow and the C-band co-polarized vertical backscatter signal ( $\sigma_0^{VV}$  hereafter). On the one hand,  $\sigma_0^{VV}$  is less sensitive to changes in snow wetness at low incidence angles (Nagler et al., 2016); on the other hand, on steep slopes, the liquid water redistributes laterally, at least partially (Wever et al., 2016). The field site of WFJ is equipped with advanced meteorological sensors recording meteorological forcings at sub-hourly resolutions, and moreover, with first snow observations dating back to 1936, it holds one of the longest recorded time series of snow measurements for a high-altitude research station (Marty and Meister, 2012). The site is ideal for intensive measurement campaigns, as it is easily accessible, protected from avalanche danger and the two huts provide shelter, storage space for instruments, power and internet connection.

The objective of the measurement campaign was to build a dataset of ground-truth reference for the interpretation of S1  $\sigma_0^{VV}$  to monitor snow melt processes. Therefore, the measurements targeted the main scattering properties of snow: temperature, density, specific surface area (SSA), liquid water content (LWC) and surface roughness. These properties needed to be measured at a high vertical and temporal resolution to track the progression of the wetting front within the snowpack, and possibly

in concomitance with S1 acquisitions. Additionally, we measured snow water equivalent (SWE), a key variable for snow melt monitoring. The resulting dataset is a time series of manually measured snow profiles describing the evolution of snow scattering properties at unprecedented vertical and temporal resolutions. The dataset consists of 38 snow profiles for the season of 2022-2023 (starting in February and ending in June) and 47 for the season of 2023-2024 (starting in November and ending in July). In dry snow conditions, measurements were carried out once per week. On the first season, once the snowpack reached the full isothermal state, measurements have been carried out regularly every second working day for a total of three times per week. On the following season, the regularity of the measurements was partially given up in favor of a better synchronization with S1 acquisitions. To get the fullest possible picture to interpret the melt dynamics, manual measurements are accompanied by automatically recorded time series of runoff and SWE.

#### 2.1 Manual measurements

#### 2.1.1 Temperature




Snow temperature serves to monitor the progression of the snowpack to the isothermal state, which allows the presence of liquid water. Profiles of snow temperature were sampled from the surface to the bottom with a vertical resolution of 10 cm on snow season 2022-2023 and of 5 cm on snow season 2023-2024 using a batch of HI98501 Checktemp from Hanna (HannaInstrumentsInc.). According to the instrument specifications, the uncertainty range is  $\pm$  0.2°C. Each temperature reading was marked down after waiting an adequate time for measurement stabilization.

#### **2.1.2 Density**

In dry snow conditions, snow density controls (i) the probability of scattering events, as denser snow has more grains per unit volume and (ii) the real part of the effective permittivity (see the following Sec. 3.2), which increases with the increased fraction of ice relative to air, typical of denser snow. Profiles of snow density were sampled from the surface to the bottom with a vertical resolution of 3 cm using a box density cutter and a digital scale. The box cutter used for this campaign has a volume of 100 cm<sup>3</sup>. The uncertainty range of this instrument is between 5 and 10% with the main sources being the presence of ice layers, the compaction of light snow while collecting the sample, or losing fractions of it in conditions of fragile snow such as facets or depth hoar (Conger and McClung, 2009; Proksch et al., 2016).

#### 2.1.3 Specific Surface Area

Snow specific surface area (SSA) expresses the surface area of snow grains per unit mass, and is related to the grain size and structure. Smaller grains give higher values of SSA – meaning that the number of scattering centers is increased, but the effect of each one is weakened. Therefore, when grains are too small, the total backscatter can decrease. Larger grains, on the other hand, give lower values of SSA – meaning that scatterers are fewer but stronger and more efficient. Therefore, with enhanced volume scattering, the overall backscatter increases. Profiles of SSA were sampled from the surface to the bottom with a vertical resolution of 4 cm using the InfraSnow sensor from FPGA (FPGA Company; Wolfsperger et al., 2022). This non-destructive

method builds upon the principle of diffuse near-infrared reflectance measurements using a compact integrating sphere setup to derive optical equivalent grain diameter (OED), and therefore SSA (Gergely et al., 2014). To compute OED, snow density is required as an input parameter and for this we use the measured density profile. With a relative error of RMSE = 15% (Wolfsperger et al., 2022) when compared to  $\mu$ -CT, this instrument seems to be slightly less accurate than others commonly used such as the IceCube (Zuanon, 2013), however, this bias is more pronounced for high values of SSA typical of dry snow, which is not the main object of our study. Moreover, the use of the InfraSnow is especially practical and portable for field applications.

#### 2.1.4 Liquid Water Content







The formation of liquid water content (LWC) in the snowpack enhances its dielectric constant, leading to higher absorption losses and significant reduction in radar penetration depth. These concepts will be addressed in more detail in Sec. 3.2. Profiles of LWC were sampled from the surface to the bottom with a vertical resolution of 2, 5 or 10 cm, depending on the method. We used dielectric sensors coupled with melting calorimetry to corroborate measurements in conditions of high LWC at later stages of the melting process. To our knowledge, this is the first time series of LWC snow profiles measured at such high vertical and temporal resolution. On the first campaign year, we used the Denoth capacitive sensor (Denoth, 1994) ("Denothmeter" hereafter). It consists of a flat capacitance probe with an estimated measurement surface of 176 cm<sup>2</sup> (Techel and Pielmeier, 2011). The probe operates at a frequency of 20 MHz and measures the real part of the permittivity of snow, and a separate measurement of density is required to obtain the imaginary part (Denoth et al., 1984; Denoth, 1989) – here, similarly than for SSA, we used the measured density profile. The Denothmeter has been widely used in field studies to monitor the evolution of snowpack wetness (Fierz and Föhn, 1994; Kattelmann and Dozier, 1999; Techel and Pielmeier, 2011), alone or in comparison with other techniques, e.g. in Koch et al. (2014); Wolfsperger et al. (2023); Barella et al. (2024). On the second campaign year, we adopted the new capacitive snow sensor (NCS hereafter) developed at the Institute for Snow and Avalanche Research SLF (Wolfsperger et al., 2023) and produced in batch series from FPGA company. The use of the Denothmeter was discontinued because it is not commercially available, and only two units were available to us, risking measurement continuity if damaged during intensive use. The NCS works in the same way as the Denothmeter, operates at the same frequency and measures over a slightly larger surface of 202 cm<sup>2</sup>. The NCS was compared against the Denothmeter in both field and laboratory settings and the agreement was generally good, however, in isolated cases of very wet layers, the measured permittivity tended to deviate towards higher values (Wolfsperger et al., 2023). A good element of consistency is that the comparison between NCS and Denothmeter was carried out within this campaign, in the snow season 2022-2023. The absolute error associated with dielectric measurements was estimated around 1% in volume (Sihvola and Tiuri, 1986; Fierz and Föhn, 1994). To our knowledge, a systematic study on the errors associated with the Denothmeter was never carried out. However, similar studies are available for the Finnish snow fork (Sihvola and Tiuri, 1986), which directly measures both real and imaginary parts of snow permittivity. The error associated to the snow fork in measuring LWC is between  $\pm 0.5\%$  (Sihvola and Tiuri, 1986) and  $\pm 0.3\%$  (Moldestad, 2005). (Techel and Pielmeier, 2011) used both the Denothmeter and the Snow Fork in their study, reporting differences of around 1% between the two instruments. Additional uncertainties for dielectric measurements derive from interference with solar radiation near the surface (Lundberg, 2008), which we tried to minimize throughout the campaign.

Because dielectric devices may lose accuracy for high LWC values (Perla and Banner, 1988; Techel and Pielmeier, 2011), for both snow seasons, in conditions of ripe snow, Denothmeter/NCS measurements were backed up with melting calorimetry following the revised field protocol recently described in Barella et al. (2024) and partially carried out within the same measurement campaign described here. This field protocol is tailored to reduce the higher uncertainty ranges previously associated to melting calorimetry (Kawashima et al., 1998; Kinar and Pomeroy, 2015; Avanzi et al., 2016). It proposes a revised formulation of the calorimetric uncertainty that incorporates the calorimetric constant and the propagation of uncertainties coming from instrument, operational and environmental conditions. The uncertainty range associated with the new protocol for melting calorimetry is  $\pm 0.5\%$  and the absolute error compared with Denothmeter measurements is  $\sim 1\%$  in volume.

#### 2.1.5 Surface Roughness






Snow surface roughness controls the scattering behavior of the snowpack surface, with smooth surfaces exhibiting a dominant specular reflection and rough surfaces behaving more similarly to a diffuse scatterer. Snow surface roughness is typically expressed using three parameters: the root mean square of the heights (RMSH), the correlation lenght (CL) and the autocorrelation function (Williams and Gallagher, 1987; Nagler and Rott, 2000; Manninen et al., 2012; Anttila et al., 2014). These parameters can be obtained from a digitized snow transect. A proven and robust system involves inserting a panel into the snow and capturing images of the snow surface with a digital camera (Manninen et al., 2012; Anttila et al., 2014). For this campaign, we used the method described in Barella et al. (2021) and refined in Barella et al. (2025), which builds upon these concepts and it is particularly suited for field applications. The panel we used is made of black Forex, 70.5 cm wide and 47 cm tall. These dimensions are a trade-off between the ease of transport and the length of the snow transect covering at least 10 times the C-band wavelength  $\lambda$ =5.5 cm as suggested in (Manninen et al., 2012). The panel can be photographed by means of any digital camera. To attain a representative snow transect, 9 pictures were taken on each measurement day: 3 along one direction, 3 along the perpendicular direction, and 3 at a 45° angle between them. The resulting roughness profile is averaged among all usable pictures, i.e., those not affected by excessive shadowing or unclean panel surface. To our knowledge, a time series of snow surface roughness properties was never measured before.

#### 2.1.6 Snow Water Equivalent

Profiles of snow water equivalent (SWE) were sampled from the surface to the bottom with a cylinder cutter of inner diameter 9.44 cm and length 55 cm. The snowpack was sampled in sections from the surface to the ground and the total SWE was obtained by weighting each sample and summing up all the values. The uncertainty range of this instrument is around 10% with the main uncertainty source being caused by the presence of ice layers (Proksch et al., 2016).

Figure 2. Overall range of local incidence angles across the study area for all the four relative orbits – morning/descending  $(M\downarrow)$  and afternoon/ascending  $(A\uparrow)$ . Each S1 cell is identified by its centroid and a number.

#### 2.2 Automatic measurements

#### **2.2.1** Runoff


Runoff was automatically measured at a sub-hourly resolution by a lysimeter. Unfortunately, the instrument was discovered to be clogged when the runoff started in 2023. The instrument was repaired only in late May 2023. Therefore, the time series for that year starts with a peak (see Fig. 8d), although we hypothesize that runoff may have started as early as the end of April 2023. To avoid similar issues, on the following season the lysimeter was inspected timely and assessed as fully functional.

#### 2.2.2 Snow Water Equivalent

Manual snow water equivalent (SWE) measurements are complemented by an automatically recorded time series at sub-hourly intervals, using the SSG1000 snow scale permanently installed at the WFJ site and manufactured by Sommer Messtechnik, Austria. The system consists of a weighing platform and load cells, which directly measure the weight of the snowpack on the platform and convert it into SWE. This instrument has a measurement range of 0 to 1000 mm of water equivalent. During the 2023-2024 snow season, the upper capacity was reached due to above-average snow depths. In comparison to manual measurements, Smith et al. (2017) estimated an error of ±10%.

#### 2.3 Sentinel-1 acquisitions

S1 is designed as a two sun-synchronous polar-orbiting satellite constellation, acquiring dual polarimetric C-band (frequency of 5.405 GHz, wavelength of 5.5 cm) SAR images with a nominal resolution up to 3.5 m × 22 m in Interferometric Wide swath mode (IW) and a revisit time of 6 days. Acquisitions in IW have a swat of approximately 250 km. This, together with the overlapping orbit paths, conceives the acquisition of multiple tracks at middle latitudes such as the Alps. For this reason, within the time window of 6 days, more acquisitions of the same area may be available. Unfortunately, Sentinel-1B failed at the end

of 2021, and with only Sentinel-1A in orbit, repeat cycles halved from 6 to 12 days. Since then, the overall data acquisition capability was reduced by ~50% in most regions, including our Weissfluhjoch field site. Data from four relative orbits are available for this site: two ascending (afternoon) and two descending (morning) passes. Figure 2 shows the overall range of local incidence angles across the field site, which vary from a minimum of 27° to a maximum of 47°. These maps highlight domains with stronger and weaker dependence on the incidence angle – an east-facing back-slope and a flat area, respectively.

The SAR images can be downloaded, free of charge, from the copernicus data hub (Copernicus). To account for the complex topography and to reduce the speckle noise of SAR acquisitions, a tailored preprocessing procedure was applied to all data. The processing procedure involves a combination of tools, some of which are available in SNAP (Sentinel Application Platform) version 6.0, while others are customized and developed in Python. The full workflow is described in Marin et al. (2020); however, in this study, the gamma-MAP filter was not applied. The final spatial resolution of the post-processed S1 images is  $20 \times 20$  m.

The nominal radiometric uncertainty of S1 falls in the range of  $3\sigma=1.0$  dB, as indicated in several ESA validation campaigns (Torres et al., 2012; Miranda et al., 2015; Schwerdt et al., 2017; Benninga et al., 2020). However, the overall radiometric accuracy is also affected by a number of preprocessing steps, including (but not limited to) the application of despeckle filters, terrain correction and radiometric normalization (particularly challenging in mountain regions with complex topography), and thermal noise removal (important in conditions of high absorption, such as wet snow). In such conditions, a detailed specification becomes extremely complex and falls beyond the scopes of this paper. Nonetheless, since this study explores the multitemporal behavior of  $\sigma_0^{VV}$  over a target cell, it is relevant to mention speckle denoising. We used the filter proposed by Quegan and Yu (2001) – a powerful yet relatively simple one to denoise multitemporal stacks, with a 11 × 11 pixels window. Similarly to local spatial multi-looking, its implementation involves local averages of intensity values for each date. Intuitively, this could potentially blur strong targets and edges, ultimately leading to a loss of resolution and impacting the overall multitemporal result. However, in conditions of dry snow, the snow cover and the position of the scatterers are stable, snow temperatures are well below 0°C and the soil should be mostly frozen, implying constrained variations in soil moisture. Under these conditions, the pixels we considered in our study exhibited an overall stable behavior. The same stability was observed during dry periods in summer. In these two cases, the standard deviation was within 1.0 dB, which aligns with the nominal radiometric uncertainty of S1. During the melting period, the primary source of radiometric uncertainty originates from the formation of LWC within the snowpack. As a consequence, the radar return signal from the same target cell changes over time, resulting in reduced temporal coherence and larger deviations in multitemporal statistics. As will be shown in the course of this study, LWC potentially exhibits high heterogeneity across a single resolution cell. Under such conditions, the estimation of radiometric uncertainty becomes particularly challenging. Without a precise reference for LWC, a rigorous uncertainty quantification is inherently difficult and lies beyond the scope of this work.

#### 2.4 Campaign design







Measurements were carried out within freshly dug snow pits, starting at 08:00 approximately. The start of the measurement procedure depended on the amount of employees available on a specific day, on the amount of snow, on its density and on

the weather conditions – generally, between one to two hours later. The measurement procedure was generally finished around 12:00 refilling the snowpit; however, on isolated days, there were several hours of delay because of the above mentioned reasons. On the first snow season, the snow temperature was generally measured first and the melting calorimetry last, with the remaining measurements being carried out in between with an order that also varied as a function of the above mentioned factors. On the second snow season, we improved the campaign design with a more rigorous measurement order: temperature first, SSA and dielectric LWC either simultaneously or one after the other, density, SWE, and melting calorimetry coupled with a second simultaneous dielectric LWC profile taken at the same vertical location. This has specific importance for the LWC profiles. On the first season, the time lag between the dielectric and calorimetric LWC profiles was 2 or 3 hours, at an horizontal distance of 50 cm to 1 m. On the second season, we measured one first dielectric LWC profile and an adjacent, simultaneous one using melting calorimetry. In Sec. 4.2, we will refer to the first setup as "co-located" and to the second one as "simultaneous".

On both seasons, before starting the measurement procedure, the profile wall was made as smooth as possible. A Near-Infra-Red picture was taken for qualitative comparison. Outside of the snow pit, on an undisturbed area, the surface roughness panel pictures were taken. On days where the radiation (from the sun or diffuse) was particularly intense, shading was necessary for every surface measurement that might have been affected. The temperature profiles were always measured in the shaded corner area of the snow pit. Overall, each measurement series would need a total horizontal space of 1.5-1.8 m, and the single variable profiles were measured at a reasonable horizontal distance from each other. On both seasons, snow profiles were carried out within the same designated area. The area was divided in corridors approximately 2 m wide. Throughout the season, measurements were carried out moving continuously forward along the corridor until the slope was hit. The next snow profile would be dug onto the next corridor. A minimum distance of 30 cm was secured between two consecutive measurement days, to avoid disturbances from the previous measurement set.

Data cleaning and homogenization procedures were performed before providing the measured snow properties as RT inputs. In particular, since sampling resolutions were different (see Sec. 2.1), all measured properties were linearly interpolated to a common vertical resolution of 1 cm. Positive LWC values recorded at temperatures below 0°C were corrected to 0%. 0.04% and 0.4% of the measured LWC values were above or equal to 15% for the two years respectively. For both dielectric instruments, these values are likely not accurate. Since these values likely represent areas of high snow wetness, they were not excluded from the analysis but their LWC value was set to 15%, similarly to Techel and Pielmeier (2011). Additionally, instances of very low LWC measurements from thin layers just above the ground in dry snow conditions were discarded, as we could not rule out potential instrument disturbances from the ground in these cases. Given the accuracy range of the thermometer (see Sec. 2.1), temperature oscillations up to 0.2°C below 0°C were set to 0°C from the first measured fully isothermal profile onwards. Since the snow properties were measured at a certain lateral distance one from the other, the profiles of density and SSA were slightly shifted with a simple algorithm to maximise the correlation with the profile of LWC. Finally, we had to discard the last 3 snowpit measurements of 2023, because the measured RMSH value there was too high to ensure the conditions of validity of the interface model (see Sec. 3.2).

Figure 3. Aerial view of the WFJ measurement station. Each of the 56 points represents the centroid of each S1 cell. Each centroid is split in two, the left part indicating the interquartile range (IQR) of the winter  $\sigma_0^{VV}$  signal for the snow season of 2022-2023 and the right part for the snow season of 2023-2024. Contour lines indicate the surrounding slopes. The yellow rectangle indicates the fenced measurement area where snow profiles were carried out in both seasons. Cell 40, i.e. the selected S1 cell for this study, is highlighted in red.

Table 1. S1 tracks overlooking the selected cell 40, with times of acquisition, direction of orbit and local incidence angles.

| Track # | Time of acquisition | Direction of orbit | Local incidence angle |
|---------|---------------------|--------------------|-----------------------|
| 015     | ~17:30              | Ascending          | 41°                   |
| 117     | ~17:30              | Ascending          | 32°                   |
| 066     | ~05:30              | Descending         | 33°                   |
| 168     | ~05:30              | Descending         | 42°                   |

#### 3 Methods and model

#### 3.1 Selection of the Sentinel-1 reference cell

The selection of the reference S1 cell required some considerations. The WFJ field site is ideal for continuous measurements due to its proximity to structures and sensors, however, these features may interfere with radar waves, thus disrupting the

Figure 4. Variability of  $\sigma_0^{VV}$  in dry snow conditions for all relative orbits overlooking cells 18, 25, 32, 38, 39, 40, i.e. the flat terrain cells with likely similar snow properties as the measured ones (a-b). Multitemporal  $\sigma_0^{VV}$  signal of the selected cell 40 compared to the ensemble standard deviation  $(std_{\sigma_0^{VV}})$  of the similar cells – morning/descending  $(M\downarrow)$  and afternoon/ascending  $(A\uparrow)$  (c-f).

backscatter from natural terrain. Most of the structures within the field site are metallic and may act as additional reflecting sources in addition to the snowpack.

To select the reference cell, we extracted  $\sigma_0^{VV}$  values for both years over a grid of 56 points covering the whole extension of the field site and the immediate surroundings (Fig. 3). For each cell and for each different year, we isolated the time frame starting at the beginning of the meteorological winter (Dec 01) and ending when the first liquid water was measured in the snowpack. Over these time frames, for each year and for each cell we computed the variability of  $\sigma_0^{VV}$  acquired by the 4 different tracks (See Tab. 1). We assume that lower variabilities between different tracks over a dry snowpack may indicate a

minimal interference with other non-natural elements on the field, as their backscatter would typically exhibit strong angular dependence (i.e., anisotropy).








The results of this analysis are shown in Fig. 3, where the variability is mapped over the field using the interquartile range (IQR). In general, the IQR does not vary significantly between the two snow seasons, suggesting that this kind of approach might be adequate to select a reference cell with the least possible artificial disturbance. Outliers – i.e., cells 15, 22, 23, 27, 52, 54, and 55 – are likely influenced by localized field conditions. These include double-bounce effects typically associated with man-made structures (e.g., cell 27), surfacing boulders (cells 52, 54, 55), or small variations in soil moisture, which could account for the observed year-to-year variability. The highest IQR values are clustered around the large hut (for double-bounce effects) and where the slopes start to become steeper (when the backscatter has strongest dependence on the incidence angle). Interestingly, the IQR values for cell 25 and 32 are among the lowest for both snow seasons, suggesting that smaller metallic sensors might not represent a significant disturbance for radar waves.

Ideally, the target cell should coincide with the location of in-situ measurements to ensure that the observed snow properties accurately represent those detected by the radar. Although S1 footprint is large relatively to the area disturbed by a single snow pit, excavating multiple consecutive snow profiles across a broader area can ultimately alter snow conditions across the entire cell – particularly under moist or wet snow conditions. This would introduce an uncontrolled degree of uncertainty. Therefore, the target cell should rather be selected among the surrounding undisturbed cells with similar slopes and aspect. Fig. 4a-b show the dry-snow  $\sigma_0^{VV}$  variability for a set of cells with such features, i.e., cells 18, 25, 38, 39, 40. Among these, cell 40 shows a distinguished dependence on each incidence angle and orbit direction, along with relatively low variability of  $\sigma_0^{VV}$ across tracks. An exception occurs for track #117 during 2023-2024, where the variability is relatively higher with respect to the year before. This increased variability is also noticeable for cell 25 and 39. Given the lower variabilities recorded on the prior year, interference from non-natural elements can be ruled out. The most plausible explanation is a certain degree of heterogeneity in soil moisture across the field. Unfortunately, we are unable to verify this hypothesis, as soil moisture measurements were not included in our field campaign. Additionally, cell 40 lies in the immediate vicinity to the measurement site, and the snow surface remains undisturbed due to the operation of a LiDAR laser scanner continuously monitoring the snow surface. Fig. 4c-f illustrates the multitemporal  $\sigma_0^{VV}$  signal from cell 40 in comparison to that of the other candidate cells. The average standard deviation of the  $\sigma_0^{VV}$  ensemble across these cells is approximately 3 dB for all tracks. Interestingly, the lowest standard deviation is consistently observed at the time of the backscatter drop caused by wet snow, with the exception of track #117 in 2024. Notably, during the melting season, the signal from cell 40 lies in the lower end of the backscatter range across all years and tracks – aside for track #117 in 2022-2023. Potentially, this behavior is desirable for wet snow detectability. For these reasons, the  $\sigma_0^{VV}$  recorded over cell 40 is selected as the reference time series for this work.

The impact of incidence angle was not a primary focus of this study, as it has already been extensively addressed in previous research Mätzler (1987); Shi and Dozier (1992); Strozzi et al. (1997); Strozzi and Matzler (1998), which strongly relied on tower-based instruments allowing greater control than satellite-based radar systems. In our case, the area most representative of measured snow properties is relatively small and flat, resulting in a limited range of local incidence angles available for analysis (see Fig. 2). Furthermore, the high spatial variability of LWC would require dedicated reference measurements for

each incidence angle and cell, which was not feasible given the time and resources already involved in conducting the campaign at a single location.

#### 3.2 Snow Microwave Radiative Transfer (SMRT) model: description and simulation setup

SMRT is a model that simulates the active-passive microwave response from layered snowpacks (Picard et al., 2018). The model is written and run in a Python environment and has a modular and flexible structure, allowing the user to set model runs choosing among a wide set of electromagnetic, microstructure and permittivity models. The reflectivity and transmissivity associated to roughness can also be described according to different models. The user has to specify a set of snowpack properties to parametrize the microstructure and the electromagnetic model. In particular, the roughness can be set either at the snow-air interface only or for each defined snow layer. Once these necessary parametrizations have been declared in the preliminary components of the model, SMRT uses the discrete ordinate and eigenvalue (DORT) method to solve the radiative transfer equation. The user can either customize a virtual sensor with specific frequency, incidence angle and polarization or directly choose from a list of already available sensors, among which S1. The backscatter intensities can be obtained for all polarizations – this study focuses on the co-polarized vertical signal, because cross-polarizations are currently only partially implemented within the current version of the module used for the parametrization of surface and interface scattering (Murfitt et al., 2024).

This study uses the symmetrized strong-contrast expansion (SymSCE) (Picard et al., 2022b) as the electromagnetic model with two different permittivity parametrizations. Measurements of density and SSA were used to compute the Porod length  $(\ell_P)$  (Porod, 1951). The microwave grain size  $(\ell_{MW})$  is computed as the product of  $\ell_P$  and the polydispersity k, a parameter describing the variability of the length scales with respect to the microstructure (Picard et al., 2022c). k was set to 0.75: this empirical value was estimated from  $\mu$ -CT scans of a wide variety of alpine snow samples with convex grains, among which rounded grains and melt forms (Picard et al., 2022c). For this study, snow microstructure was parametrized using the exponential model. For frequencies in the X- and Ku-bands (10-17 GHz), exponential auto-correlation functions have been shown to be too simplistic for representing snow microstructure. Their fast decay fails to capture long-range spatial correlations, and their inadequacy in modelling densely clustered media results in an underestimation of forward scattering effects (Chang et al., 2016). However, Picard et al. (2022c) show how  $\ell_{MW}$  can be computed analytically for various forms of auto-correlation functions, including the exponential. These analytical expressions of  $\ell_{MW}$  allow for direct comparison between different representations of snow microstructure. Most importantly, when the same value of  $\ell_{MW}$  is used as input, all microstructure models give the same scattering amplitude in the low-frequency limit. Therefore, according to these findings, the choice of the best representation of snow microstructure becomes a secondary problem with respect to measuring  $\ell_{MW}$  in order to predict snow scattering in the C-band.

The permittivity of a material is a complex number composed of a real part (i.e., the dielectric constant) and an imaginary part. The contribution of the real part is related to the material's ability to store electrical energy, whereas the contribution of the imaginary part is associated with dielectric losses. Snow is a three-component mixture of ice, air and water – therefore, the effective permittivity of snow  $(\epsilon_s)$  depends on the relative proportions of these elements. The presence of liquid water

Figure 5. Real and imaginary parts of the effective permittivity ( $\epsilon_s$ ) of wet snow as a function of frequency (f) for a nominal density value of 400 kgm<sup>-3</sup> and varying LWC of 1% (a), 4% (b) and 8% (c) according to the MEMLSv3 and H-86 permittivity models. Grey dotted lines underline differences between the formulations at the nominal frequency of S1, i.e. 5.405 GHz.





significantly alters both the real and imaginary parts of  $\epsilon_s$ , affecting how microwaves interact with the snowpack. Henceforth, accurate estimates of  $\epsilon_s$  are crucial for interpreting the microwave response of wet snow. Despite extensive research, particularly in the 1980s, a universally accepted model for snow permittivity has not yet been established (Picard et al., 2022a). For this study, we selected two formulations: (i) the Microwave Emission Model for Layered Snowpacks (Wiesmann and Mätzler, 1999) in its 3<sup>rd</sup> version (MEMLSy3 hereafter), which is based on the Maxwell-Garnett mixing theory of dry snow and prolate water inclusions; (ii) the Debye-like model modified by Hallikainen et al. (1986) (H-86 hereafter), which uses a mixing formula based on volume fractions and refractive indices, calibrated against field data. These models were selected because they were validated against real-world C-band data. Specifically, in Hallikainen et al. (1986) and earlier works, the authors present what is, to our knowledge, the only available dataset of wet snow permittivity measurements at 6 GHz for varying LWC values, measured using freezing calorimetry. Interestingly, MEMLSv3 fails to accurately reproduce this dataset. However, Kendra et al. (1998) observed that the dielectric constant provided by H-86 appears to be too low, an observation that is supported by data from Achammer and Denoth (1994), collected in the range between 8 and 12 GHz. However, these data appear to favor H-86 over MEMLSv3 when considering the imaginary part of  $\epsilon_s$ . While H-86 has been criticized, some aspects appear to have been overlooked (e.g. the recent corrigendum in Picard et al. (2022a)). Figure 5 shows the real and imaginary parts of the  $\epsilon_s$  as a function of the frequency for a nominal density value and varying values of LWC according to both MEMLSv3 and H-86 permittivity formulations. For higher values of LWC (see Fig. 5b,c), the  $\epsilon_s$  values obtained from both formulations display a frequency dependence and curve shape closely resembling that of pure water. In both cases, the real part of  $\epsilon_s$  decreases with frequency, whereas the imaginary part increases up to the relaxation frequency and decreases thereafter. However, in the C-band, the two formulations diverge significantly, especially in their prediction of the imaginary part, which governs absorption losses. This difference becomes more pronounced for increasing values of LWC. For instance, at LWC=4%,

MEMLSv3 predicts an imaginary part of  $\epsilon_s$  approximately twice that of the H-86 at the nominal frequency of S1 (see Fig. 5b). Since we cannot definitively determine the fitness of one model over the other, both formulations will be used in SMRT for this study. Given the different behavior of the two formulations, we expect a lower and upper bound for S1 backscatter simulations. It is clear that further research is needed to accurately characterize wet snow permittivity, but this is out of the scope of this paper.







RT modelling of snow comes with the additional difficulty of quantifying the dense medium effects, i.e., the electromagnetic interactions occurring between snow grains that are closely packed together. At C-band frequencies, these effects become significant as the scattering regime changes due to the presence of liquid water – both through changes in snow grain interactions and in bulk dielectric properties. In H-86, dense medium effects are not accounted for. In MEMLSv3, these effects are accounted through a semi-empirical parametrization involving, among other parameters, correlation length, density-dependent corrections and – as mentioned above – mixing formulas. Correlation lengths are used to represent the effective grain size and spatial correlation of the ice matrix, and to capture the degree of interaction between dense grains. Despite the range of correlation lengths being limited in MEMLSv3, the ones that are represented derive from structures observed at Weissfluhjoch during two snow seasons (Wiesmann and Mätzler, 1999). Therefore, they are likely suitable to describe the dense medium effects on the snowpack structures observed and measured in this study. Snow density is used as a proxy to determine how closely grains are packed; and as density increases, scattering is reduced and absorption increases. Such corrections are embedded into the extinction term, i.e., the sum of scattering and absorption coefficients.

The chosen interface model (between snow and air and between snow layers) is the integral equation model (IEM) (Brogioni et al., 2010), since it is one of the most used models to describe the roughness. However, any other model could be used, provided the roughness characteristics are within the validity range. The IEM is valid under the conditions  $w \cdot RMSH 

**Figure 6.** (a) Vertical profiles of snowpack properties measured in the field on May 14, 2024: temperature (dark red), density (dark yellow), liquid water content (LWC; light blue), and specific surface area (SSA; dark blue). The vertical spacing of the points connected by the lines reflects the measurement resolution for each profile: 5 cm for temperature, 3 cm for density, 2 cm for LWC, and 4 cm for SSA. (b) Representation of the same profiles averaged according to the physically consistent snow layers (indicated by grey horizontal lines). The layered profiles as in (b) form the input snowpack for the SMRT model, combined with surface roughness parameters measured on the same day (RMSH=2.7 mm; CL=48.5 mm).

was more heterogeneous, therefore requiring more layers in the model to remain as true as possible to the conditions observed in the field. Despite the efforts to find a reasonable compromise between all the above mentioned constraints, the optimal way to model a radar-equivalent snowpack from field measurements and/or detailed multilayer physical model outputs remains an

500

**Figure 7.** Variability of the number of modelling layers in SMRT used for each simulation day as a function of the melting phase and the campaign year.

open question in the field of radiative transfer modelling of snow, only recently addressed by Meloche et al. (2024), albeit for dry snow only.

### 4 Results

505

510

# 4.1 Identification and re-definition of melting phases from multitemporal Sentinel-1 backscatter and field measurements

Fig. 8-9 show the evolution of the multitemporal S1 SAR backscatter together with the time series of measured properties: snow temperature, LWC, air temperature, total water content (TWC), runoff, snow water equivalent (SWE) and surface roughness indices (RMSH and CL). The melting phases identified with the method proposed by Marin et al. (2020) are reported on each time series for later validation. We will refer to the snow seasons of 2022-2023 and 2023-2024 as the 2023 and 2024 seasons, respectively.

Our roughness measurements show clear differences for different snow surfaces (Fig. 10). Smooth surfaces typical of new/dry snow have RMSH values around 1 mm (Fig. 10a). Thereon, roughness increases with increasing surface degradation due to melt-refreeze cycles and sublimation (Fig. 10b). The values of RMSH measured in these conditions, which are the most persistent throughout the melt season, lie within 3 and 10 mm approximately. Fully-formed suncups are associated to values of RMSH around 10-15 mm (Fig. 10c). Deep suncups appear like craters on the snow surface (Fig. 10d), some reaching

Figure 8. Data overview for the snow season of 2022-2023. (a) S1 backscatter time series: exact values of  $\sigma_0^{VV}$  acquisitions (triangles); range obtained by connecting the consecutive S1 passages by direction of orbits, i.e. by connecting all the morning/descending ( $M\downarrow$ ) and the afternoon/ascending ( $A\uparrow$ ) acquisitions (shaded areas). Each panel is subdivided into the melting phases identified with the method of Marin et al. (2020). (b) Manually measured profiles of snow temperature. (c) Manually measured profiles of snow liquid water content (LWC). (d) Air temperature at hourly resolution, measured by the automatic sensor at WFJ. (e) Measured total water content (TWC) (light blue); runoff time series automatically recorded by the lysimeter at WFJ (dark blue); lack of runoff data due to the instrument failure (grey area); snow water equivalent (SWE) both automatically recorded by the snow scale (black line) and manually measured (white circles). (e) Time series of measured surface roughness parameters – RMSH and CL.

width of 20 cm and depths of 10 cm. In these conditions, we measured values of roughness RMSH equal or higher than 20 mm.

**Figure 9.** Data overview for the snow season of 2023-2024. (a) S1 backscatter time series: exact values of  $\sigma_0^{VV}$  acquisitions (triangles); range obtained by connecting the consecutive S1 passages by direction of orbits, i.e. by connecting all the morning/descending and the afternoon/ascending acquisitions (shaded areas). Each panel is subdivided into the melting phases identified with the method of Marin et al. (2020). (b) Manually measured profiles of snow temperature. (c) Manually measured profiles of snow liquid water content (LWC). (d) Air temperature at hourly resolution, measured by the automatic sensor at WFJ. (e) Measured total water content (TWC) (light blue); runoff time series automatically recorded by the lysimeter at WFJ (dark blue); snow water equivalent (SWE) both automatically recorded by the snow scale (black line) and manually measured (white circles). (e) Time series of measured surface roughness parameters – RMSH and CL.

In 2023, the first liquid water was measured on Apr 10 (Fig. 8b). On this date, our data show that the temperature of the top  $\sim$ 5 cm of the snowpack was 0°C (Fig. 8b). The air temperature reached 0°C as well on this day (Fig. 8d). The snowpack reached full isothermal state 20 days later. Ice layers formed throughout the season, likely as a consequence of repeated melt-

**Figure 10.** The panels illustrate some representative surface roughness conditions as qualitatively observed on the field (panoramic pictures) together with one of the panel measurements performed on the same day (bottom right of each panel, where the mean roughness RMSH measured on that day is also reported). (a) Smooth surface typical of dry snowpack conditions. (b) Early-stage development of surface roughness deriving from melt-refreeze cycles. (c) Fully-formed suncups over a homogeneous snow cover, at least among the considered S1 cell. (d) Fully-formed suncups over a patchy snow cover.

refreeze cycles and the succession of several warm and cold spells (Fig. 8d). Ice layers were observed regularly during the measurement campaign, their presence is highlighted by locally higher values of LWC due to ponding at approximately 100 cm from the ground. The presence of ice layers probably withheld the meltwater in the upper section of the snowpack, partially hindering the progression of the wetting front. LWC profiles in Fig. 8c highlight ponding above ice layers consistently until May 15. The ponding is no longer detected over the next consecutive 5 snow profiles and becomes visible again from May 26 until early June, when the ice layers likely disintegrated allowing the meltwater to percolate to the bottom of the snowpack. The fact that the ponding above ice layers is not detected on a series of consecutive snow profiles is probably linked to the partial refreeze of the snowpack highlighted by the drop in air temperature detected within this time span (Fig. 8d). However, ice layers could also be laterally non homogeneous. Fig. 8f shows that the roughness associated with wet snow starts developing short after the snowpack starts moistening, with RMSH increasing until May 9. Thereon, the cold spell brought new snowfalls which smoothened the snow surface significantly, and roughness indices reverted to typically winter values for approximately 10 days. Fully-formed suncups were observed on the field from May 31 onwards. As explained in Sec. 2.2, the lysimeter time series for 2023 (Fig. 8e) is not useful to detect the runoff start. However, the automatic measurements indicate the first slight SWE decrease around May 8, following a warm spell that lasted several days. This occurred in the presence of a fully isothermal snowpack, suggesting that meltwater may have started to be released around this time.

In 2024, the first liquid water on the surface was measured on Apr 8 during a warm spell (Fig. 9b-d). From this date on, the wetting front moved somewhat into the snow before being interrupted by a cold spell, which caused a partial surface refreeze. The snowpack reached the full isothermal state on May 9. Over the course of this season, ice layers were not observed in the field, the progression of the wetting front was not hindered and the snowpack reached full saturation earlier with respect to the previous year. The runoff time series confirms that the snowpack released the first meltwater around Apr 8 – on this date, the (point-wise) measurements show a largely isothermal snowpack. Likely, the snowpack was isothermal over the entire cell (Fig. 9b,c,e). Additionally, significant amounts of LWC were measured at the ground interface after Apr 8 and the manual measurements show a SWE decrease of  $\simeq 100$  mm between Apr 4 and 15. These observations can validate the same hypothesis made for the previous season in the absence of runoff data due to instrument failure. Our measurements in Fig. 9f show that surface roughness increased relatively late (Jun 3) with respect to the previous season, with fully-formed suncups being visible on the field from Jun 19 onwards.

Coupling the detailed, high temporal resolution information about the state of the snowpack with the multitemporal SAR  $\sigma_0^{VV}$  recorded by S1 on morning and afternoon overpasses (Fig. 8-9a) enables the validation of the methodology proposed by Marin et al. (2020) to identify the melting phases. According to the authors, a drop of at least 2 dB with respect to the winter mean in the afternoon/ascending  $\sigma_0^{VV}$  identifies the start of the moistening phase; the ripening phase starts when the morning/descending  $\sigma_0^{VV}$  signal shows the same drop of at least 2 dB; the runoff starts when both morning/descending and afternoon/ascending  $\sigma_0^{VV}$  time series reach their local minima before the monotonic increase (the authors propose an average date between the two local minima when both the S1 satellites were available). For the two seasons, we computed the average winter backscatter ( $\overline{\sigma_0^{VV}}$ ) by averaging all values recorded by each individual track over the course of the meteorological

Table 2. Overview on the identification of the melting phases based on the multitemporal S1 SAR backscatter as proposed by Marin et al. (2020). For each season, the table shows the relevant values of  $\sigma_0^{VV}$  and the occurrence dates for each afternoon/ascending (A $\uparrow$ ) and morning/descending (M $\downarrow$ ) look (and corresponding incidence angle). The selected values for the start of the moistening, ripening and runoff phases are highlighted in bold. For the runoff start, the selected date according to the method of Marin et al. (2020) is compared against the data recorded by the lysimeter, when available.

| Season                                 | 2022-2023             |          |          | 2023-2024 |                        |          |          |          |
|----------------------------------------|-----------------------|----------|----------|-----------|------------------------|----------|----------|----------|
| Track                                  | 015 (A†)              | 117 (A†) | 066 (M↓) | 168 (M↓)  | 015 (A↑)               | 117 (A↑) | 066 (M↓) | 168 (M↓) |
| Local Incidence Angle                  | 41°                   | 32°      | 33°      | 42°       | 41°                    | 32°      | 33°      | 42°      |
| $\overline{\sigma^{VV}_{0,dry}}$ [dB]  | -12.3                 | -11.4    | -8.4     | -10.0     | -12.6                  | -11.5    | -8.9     | -10.1    |
| Moistening start date                  | Apr 22                | Apr 29   | _        | -         | <b>Apr 04</b> – Apr 16 | Mar 18   | -        | -        |
| Moistening start value [dB]            | -18.5                 | -16.3    | -        | -         | -14.1 – -20.0          | -13.9    | -        | -        |
| Ripening start date                    | _                     | _        | Apr 26   | Mar 28    | _                      | -        | Apr 08   | Apr 15   |
| Ripening start value [dB]              | _                     | _        | -12.6    | -13.3     | _                      | -        | -12.8    | -17.9    |
| $\sigma^{VV}_{0,min}$ , date           | May 16                | Apr 29   | May 08   | May 03    | May 22                 | May 17   | May 26   | Jun 02   |
| $\sigma^{VV}_{0,min}$ , value [dB]     | -21.4                 | -16.3    | -19.8    | -22.4     | -22.6                  | -23.7    | -20.7    | -22.8    |
| Runoff start date (Marin et al., 2020) | May 06                |          |          | May 24    |                        |          |          |          |
| Runoff start date (Lysimeter)          | No data – ∼Apr 29 (?) |          |          | ∼Apr 15   |                        |          |          |          |

winter, i.e., from Dec 01 to Feb 28. The resulting values are the benchmark needed to identify the melting phases. The results are listed in Tab. 2. As noted by Marin et al. (2020), the dependence of  $\sigma_0^{VV}$  on incident angles remains as a residual effect.

Because for the selected cell two morning/descending and afternoon/ascending looks are available, there are two possible dates for the start of the moistening and ripening phase, respectively. In 2023, these dates are Apr 22 and 29 for the moistening phase and Mar 28 and Apr 26 for the ripening phase. For the start of the moistening phase, we selected the earliest, i.e. Apr 22. For the start of the ripening phase, the two identified dates are almost one month apart, however, the  $\sigma_0^{VV}$  decrease recorded on Mar 28 by track #168 derives from a melt-refreeze cycle, as the following value recorded by the same track aligns back around the winter mean. Therefore, we selected Apr 26 as the start of the ripening phase. In 2024, for the moistening phase, the  $\sigma_0^{VV}$  value recorded on Apr 04 by track #015 is only 1.5 dB lower than  $\overline{\sigma_{0,dry}^{VV}}$ , however, the next passage of the same track on Apr 16 recorded a drop of already 7.4 dB. Therefore, the moistening start has been placed on Apr 04. On this date, track #117 recorded a drop of 7 dB with respect to  $\overline{\sigma_{0,dry}^{VV}}$ . For the ripening start, we chose Apr 15.



These considerations show that the method of Marin et al. (2020) is limited by the the halved S1 revisit frequency. This becomes even more clear for the selection of the runoff start date, as the wider separation between local minima of  $\sigma_0^{VV}$  considering all 4 looks is 17 days for 2023 and 16 days for 2024. Using the date in between to determine the runoff start, as done by Marin et al. (2020), gives potentially unreliable results in these conditions. This low temporal resolution makes it difficult to pinpoint precise onset dates, especially when minima are separated by such long periods. Despite the ambiguities, on both seasons, the identified moistening phase coincides exactly with the first snowpack warming and the consequent formation of liquid water. The identified ripening phase is also mostly consistent with the theory, as field measurements show that the

**Figure 11.** Bias between LWC measurements with dielectric devices and melting calorimetry for snow seasons of 2023 (a) and 2024 (b). In 2024, direct comparisons between simultaneous (brown) and co-located (light blue) measurements were also performed.

snowpack transitions to the fully isothermal state with the wetting front progressing to the bottom, although this process is partially hindered in 2023 by ice layers. In 2024, a sudden cold spell at the beginning of the ripening phase caused the refreezing of the superficial meltwater (Fig. 9b-d). This generated a sharp increase in both morning and afternoon  $\sigma_0^{VV}$  (Fig. 9a). In 2024, the first instance of measuring a fully isothermal snowpack coincided precisely with the first afternoon local minimum of  $\sigma_0^{VV}$ . The same cannot be verified for 2023, which instead shows a counterintuitive case where the local minimum of morning  $\sigma_0^{VV}$  anticipates the local minimum of afternoon  $\sigma_0^{VV}$  (Fig. 8a). Nonetheless, by the time the morning  $\sigma_0^{VV}$  reached its local minimum in 2023, the snowpack had already been fully isothermal for at least 5 days (Fig. 8a-b). This suggests that the snowpack is likely to be fully isothermal when the afternoon  $\sigma_0^{VV}$  reaches its local minimum. The runoff time series in 2024 shows that the snowpack had started to release meltwater as soon as in the late moistening phase (Fig. 9e), in correspondence of the first local minimum of the afternoon  $\sigma_0^{VV}$  time series on Apr 16 (Fig. 9a).

Marin et al. (2020) proposed three possible explanations for the monotonic backscatter increase following the local minima: (i) the increase in surface roughness, (ii) the decrease in TWC and (iii) the snow cover gradually becoming patchy. Our data show that over a high-altitude alpine snowpack like the study plot at WFJ, roughness develops on the snow surface well before the snow cover begins to disappear in patches. Therefore, at least for similar altitudes, the gradual disappearance of the snow cover can be ruled out as a cause of the increasing backscatter in the late melting stage. For both seasons, our data indicate that the strongest correlation with the monotonic increase of  $\sigma_0^{VV}$  after the local minimum is observed with the gradual increase in surface roughness (Fig. 8-9f). Conversely, there seems to be no remarkable correlations between the increase in  $\sigma_0^{VV}$  and the TWC and/or runoff trends. In fact, Fig. 8-9e show that the decrease of TWC as a consequence of snow ablation is not monotonous. On the other hand, both automatic and manual measurements show that by the time SWE started decreasing monotonically (around May 26, 2023 and Jun 06, 2024), the S1  $\sigma_0^{VV}$  had already increased again by  $\simeq$ 6 dB.

## 4.2 Instrumental uncertainty and variability in field measurements of liquid water content





Fig. 5 shows that liquid water has a strong impact on the real and imaginary parts of  $\epsilon_s$  at C-band frequencies. For S1  $\sigma_0^{VV}$  retrievals from ground measurements, this poses three major challenges. In the first place, manual measurements concern a

very small area/volume whereas satellite acquisitions cover a pixel size of  $20 \times 20$  m. Secondly, the distribution of liquid water within the snowpack can be highly heterogeneous because of a variety of features and processes, namely capillary barriers, preferential flows, ice layers. Finally, what is the most accurate methodology for measuring LWC in both lab and field environments remains a debated question in snow science (Barella et al., 2024), and although the methods used in this paper were designed to achieve a good level of robustness, they are nevertheless subject to error. Therefore, all these uncertainty sources need to be taken into account when comparing satellite  $\sigma_0^{VV}$  signatures with retrievals driven by measured data.






In Sec. 2.4, we explained how dielectric measurements were validated against melting calorimetry in conditions of ripe snow. We referred to the validation setup of 2023 as "co-located" only; whereas in 2024 we performed an additional "simultaneous" validation in addition to the co-located. Figure 11 shows the spread between dielectric and calorimetric measurements in co-located and simultaneous setups for all the LWC validation measurements made over the two years. In 2023, the average maximum bias between co-located measurements is 2.6% and the average standard deviation is 1.2%. In 2024, the average maximum bias and the average standard deviation are 2.6% and 1.4% for co-located measurements and 2.3% and 1.5% for simultaneous measurements, respectively. Figure 13 shows all the measured vertical profiles in detail. In 2023, there is an overall good agreement between dielectric and calorimetric measurements. The time lag between the measurements is highlighted by often similar LWC profile shapes, with calorimetry generally measuring higher peak values. Unexpectedly, in 2024, the simultaneous measurements resulted in only slightly lower biases and slightly higher standard deviations. This counterintuitive result is supported by a number of previous studies. For example, Donahue et al. (2022) found an average standard deviation of 1% over 10 cm wide snow samples with LWC between 0 and 5%. The study of Techel and Pielmeier (2011) confirms the high occurrence of measurement deviations of more than 1% at short horizontal distances. However, Techel and Pielmeier (2011) also show that the correlation between measurements at larger horizontal distances is higher for LWC values lower than 1.3%. Therefore, the biases and standard deviations observed in our field measurements may overestimate the instrument uncertainty and/or variability over larger scales comparable to the footprint of S1. Based on these considerations, we define the large-scale LWC variability as  $\pm 1\%$ . We use this value to assess the effect of LWC uncertainty on  $\sigma_0^{VV}$  retrievals from ground measurements.

# 4.3 Interpretation of Sentinel-1 backscatter through SMRT simulations forced by field measurements

Fig. 12 shows the comparison between the time series of S1 acquisitions and SMRT-modelled  $\sigma_0^{VV}$  forced by snowpit measurements using the two different permittivity formulations (MEMLSv3 and H-86) and the model setup described in Sec. 3.2, considering the LWC variability of  $\pm 1\%$  estimated in Sec. 4.2. In this Figure, together with Tab. 3, simulation results are categorized into groups, and potential sources of inconsistencies and/or driving scattering mechanisms are discussed for each group, based on the measured values of LWC, TWC, and surface roughness. All measured profiles of LWC, along with the corresponding TWC and RMSH values, are presented in Fig. 13 and Tab. 5, which serve as a reference for the following analysis. Tab. 4 shows all the Root Mean Squared Errors (RMSE) between S1 acquisitions and simulations, according to the snow season, the selected permittivity formulation and the melting phase. In general, both models exhibit a mean negative bias of  $\simeq$ 5 dB with respect to S1 recordings over both seasons; however, biases are more pronounced for 2024 than for 2023, with

**Table 3.** Supplementary information to Fig. 12: measured values of TWC, LWC, RMSH, noteworthy events for scattering (such as cold spells or late snowfalls), and explanations to the mismatch between modelled and recorded S1 backscatter signatures.

| Group      | TWC                 | LWC | RMSH   | Event                                                              | Source(s) of inconsistency, scattering mechanism                                                                                                                                                          |
|------------|---------------------|-----|--------|--------------------------------------------------------------------|-----------------------------------------------------------------------------------------------------------------------------------------------------------------------------------------------------------|
| 1a         | -                   | _   | -      | Soil thawing                                                       | - Backscattering increase due to soil thawing                                                                                                                                                             |
| 2a         | <10 mm              | <3% | 1 mm   | Snowpack moistening Smooth surface                                 | <ul> <li>Uncertainty in spatiotemporal LWC/TWC</li> <li>Scattering from surface structures (melt-refreeze)</li> <li>Surface roughness underestimation</li> <li>Wet soil scattering</li> </ul>             |
| 3a         | >10 mm              | >3% | 1→4 mm | Snowpack ripening Formation of surface roughness                   | <ul> <li>Uncertainty in spatiotemporal LWC/TWC</li> <li>Uncertainty in surface roughness measurements</li> <li>Uncertainty in IEM modelling</li> </ul>                                                    |
| 4a         | >10 mm              | >3% | 3∼4 mm | Snowpack ripening Increasing surface roughness                     | <ul><li>Uncertainty in spatiotemporal LWC/TWC</li><li>Uncertainty in surface roughness measurements</li><li>Uncertainty in IEM modelling</li></ul>                                                        |
| 5a         | >10 mm              | >3% | ~1 mm  | New snowfall on a wet snowpack Well-developed surface roughness    | - "Buried surface roughness"                                                                                                                                                                              |
| 6a         | <10 mm              | <3% | ~1 mm  | Cold spell (partial snowpack refreeze) Smooth surface              | <ul> <li>Uncertainty in spatiotemporal LWC/TWC</li> <li>Scattering from surface structures (melt-refreeze)</li> <li>Uncertainty in surface roughness measurements</li> <li>Wet soil scattering</li> </ul> |
| 7a         | >10 mm              | >3% | >4 mm  | Wet snowpack Fully-formed suncups                                  | - Uncertainty in spatiotemporal LWC/TWC - Uncertainty in surface roughness measurements - Uncertainty in IEM modelling                                                                                    |
| 1b         | <10 mm              | <3% | ~1 mm  | Snowpack moistening<br>Smooth surface                              | <ul> <li>Uncertainty in spatiotemporal LWC/TWC</li> <li>Scattering from surface structures (melt-refreeze)</li> <li>Surface roughness underestimation</li> <li>Wet soil scattering</li> </ul>             |
| 2 <i>b</i> | >10 mm              | >3% | ~1 mm  | Snowpack moistening Smooth surface                                 | <ul> <li>Uncertainty in spatiotemporal LWC/TWC</li> <li>Scattering from surface structures (melt-refreeze)</li> <li>Surface roughness underestimation</li> </ul>                                          |
| 3b         | <10 mm<br>(Varying) | <3% | ~1 mm  | Cold spell (partial snowpack refreeze) Smooth surface              | <ul> <li>Uncertainty in spatiotemporal LWC/TWC</li> <li>Scattering from surface structures (melt-refreeze)</li> <li>Surface roughness underestimation</li> <li>Wet soil scattering</li> </ul>             |
| 4b         | >10 mm              | >3% | ~1 mm  | Snowpack ripening Smooth surface                                   | <ul> <li>Uncertainty in spatiotemporal LWC/TWC</li> <li>Scattering from surface structures (melt-refreeze)</li> <li>Surface roughness underestimation</li> </ul>                                          |
| 5b         | >10 mm              | >3% | ~3 mm  | Snowpack ripening Increasing surface roughness                     | <ul> <li>Uncertainty in spatiotemporal LWC/TWC</li> <li>Uncertainty in surface roughness measurements</li> <li>Uncertainty in IEM modelling</li> </ul>                                                    |
| 6b         | >10 mm              | >3% | ~1 mm  | New snowfall on a wet snowpack<br>Well-developed surface roughness | - "Buried surface roughness"                                                                                                                                                                              |
| 7b         | >10 mm              | >3% | >4 mm  | Wet snowpack Fully-formed suncups                                  | - Uncertainty in spatiotemporal LWC/TWC  - Uncertainty in surface roughness measurements  - Uncertainty in IEM modelling                                                                                  |

Figure 12. Comparison between the recorded multitemporal S1  $\sigma_0^{VV}$  (triangles and shaded areas) and the time series of  $\sigma_0^{VV}$  modelled with SMRT, for year 2023 (a) and 2024 (b). Results are shown for both permittivity formulations – MEMLSv3 (dark gray boxplots) and H-86 (light gray boxplots). The boxplots indicate the variability associated to the LWC uncertainty of  $\pm 1\%$  for each layer, as discussed in Sec. 4.2. The shaded areas of the recorded S1 multitemporal  $\sigma_0^{VV}$  represent the range of values obtained by connecting the consecutive passages by direction of orbits, i.e. by connecting all the morning/descending and the afternoon/ascending acquisitions. The triangles represent the exact values of the acquisitions. For clarity, exact values are only shown for days where snow measurements were carried out, thus allowing direct comparison. Colored boxes group similar simulation results and are labeled with codes (e.g., 1a, 2a), which refer to Tab. 3 for details on the corresponding measured snow properties, dominant scattering mechanisms, and potential sources of error. At the top of each panel, the time series are further segmented into the melting phases identified in Sec. 4.1 – as well as the main scattering regimes, which are influenced by LWC, surface roughness, and buried surface roughness.

the deviation between permittivity models being higher as well in 2024. H-86 generally gives higher  $\sigma_0^{VV}$  values with respect to MEMLSv3.

**Table 4.** RMSE (in dB) between modelled and recorded  $\sigma_0^{VV}$  values according to the snow season, the selected permittivity formulation and melting phase.

| Season                   |                        | 2022-2023 | 3                   | 2023-2024 |              |                     |  |
|--------------------------|------------------------|-----------|---------------------|-----------|--------------|---------------------|--|
| Permittivity formulation | H-86 [dB] MEMLSv3 [dB] |           | Data to compare [#] | H-86 [dB] | MEMLSv3 [dB] | Data to compare [#] |  |
| Overall                  | 3.4                    | 4.5       | 9                   | 6.2       | 7.5          | 28                  |  |
| Dry                      | 0.5                    | 0.7       | 5                   | 0.7       | 0.5          | 4                   |  |
| Moistening               | _                      | _         | 0                   | 9.1       | 12.2         | 3                   |  |
| Ripening                 | 5.8                    | 7.6       | 3                   | 8.4       | 10.1         | 10                  |  |
| Runoff                   | 0.2                    | 1.9       | 1                   | 3.3       | 2.7          | 11                  |  |

In 2023, the #066 morning S1 track recorded a backscatter increase of more than 2 dB between Apr 05 and 19. Similarly, in 2024, we observe a 2.5 dB increase in backscatter recorded track #117 from Feb 08 to Mar 04. We can hypothesize that such increases are driven by the thawing of the soil. However, our data are insufficient and too uncertain to prove so, because of possible interferences between dielectric instruments and the ground in mostly dry snow conditions, as mentioned in Sec. 2.1.4. In dry snow conditions, there were no significant discrepancies between S1 and simulations; henceforth Fig. 12 only focuses on the period after the assumed soil thawing.





Aside the chosen permittivity formulation, five primary sources of uncertainty may account for the differences between simulated and recorded  $\sigma_0^{VV}$ . A significant one is snow transformation and melting between satellite and measurement acquisitions. S1 orbits intersect the field area either in the early morning or in the late afternoon (see Tab. 1). As explained in Sec. 2.4, measurements started at around 10:00 and would take several hours. Thus, it is likely that in both cases the LWC during the passage is lower than the value measured at 10:00 or later because of daily melt-refreeze cycles, especially near the snowpack surface. Moreover, the point-wise LWC measurements are not necessarily representative of the general liquid water distribution over the entire S1 cell. In 2023, we consistently observed ice layers over a high number of consecutive snow profiles (see Fig. 8c and 13). Our consecutive measurements suggest that ice layers contributed creating a more spatially homogeneous liquid water distribution by acting as a natural drainage barrier for meltwater. Unlike 2023, in 2024 ice layers were not consistently observed in the field. Likely, the melting process was more heterogeneous over the S1 cell, and point-wise measurements are less representative of wider scales in this season. This explains the fact that days marked by high variability associated with LWC are more numerous in 2024 than in 2023. In Tab. 3, we grouped these sources of uncertainties together under the labels "uncertainty in spatiotemporal LWC/TWC". Potentially, this source of uncertainty affects every S1 retrieval from field data. However, it definitely carries more weight than other sources of error at early melt stages when the simulation variability associated to LWC uncertainty is particularly high, i.e., when the TWC is low (Apr 24-26, 2023; Apr 04, 2024) and during both the cold spells of 2023 (May 17) and 2024 (Apr 15-23), which caused the partial refreeze of the snowpack (see Fig. 8, 9b-d).

Daily melt-refreeze cycles, however, not only alter the amount of LWC/TWC in the snowpack, but also drive the formation of surface structures that can create additional scattering which is not accounted for in the simulations, i.e. crusts (Lund et al., 2022; Brangers et al., 2024). In Tab. 3, we labeled this uncertainty source as "scattering from surface structures (melt-refreeze)".

Table 5. Total water content (TWC) and surface roughness (RMSH) values measured for the LWC profiles shown in Fig. 13.

|        | 2023 |      | 2024   |      |      |  |  |
|--------|------|------|--------|------|------|--|--|
| Date   | TWC  | RMSH | Date   | TWC  | RMSH |  |  |
| Date   | [mm] | [mm] | Date   | [mm] | [mm] |  |  |
| Apr 24 | 3    | 3    | Apr 04 | 2    | 1    |  |  |
| Apr 26 | 1    | 1    | Apr 08 | 13   | 2    |  |  |
| Apr 29 | 113  | 1    | Apr 11 | 14   | 1    |  |  |
| May 01 | 39   | 2    | Apr 15 | 34   | 2    |  |  |
| May 03 | 39   | 2    | Apr 16 | 6    | _    |  |  |
| May 05 | 114  | 3    | Apr 18 | 3    | 1    |  |  |
| May 08 | 143  | 3    | Apr 23 | 4    | 2    |  |  |
| May 09 | 102  | 4    | Apr 27 | 11   | 1    |  |  |
| May 11 | 18   | 1    | May 02 | 16   | 2    |  |  |
| May 12 | 22   | 2    | May 09 | 161  | 1    |  |  |
| May 15 | 14   | 1    | May 10 | 62   | 2    |  |  |
| May 17 | 11   | 1    | May 14 | 46   | 3    |  |  |
| May 19 | 36   | 3    | May 21 | 96   | 1    |  |  |
| May 22 | 72   | _    | May 22 | 110  | 1    |  |  |
| May 24 | 24   | 3    | May 29 | 80   | 2    |  |  |
| May 26 | 129  | _    | Jun 03 | 145  | 4    |  |  |
| May 29 | 116  | 4    | Jun 07 | 115  | 3    |  |  |
| May 31 | 193  | 7    | Jun 10 | 44   | 4    |  |  |
| Jun 02 | 27   | 10   | Jun 14 | 47   | 6    |  |  |
| Jun 05 | 38   | _    | Jun 19 | 129  | 14   |  |  |
| Jun 07 | 67   | 9    | Jun 22 | 71   | 12   |  |  |
| Jun 09 | 98   | 16   | Jun 26 | 63   | _    |  |  |
| Jun 12 | 16   | 16   | Jun 27 | 42   | 12   |  |  |
| Jun 14 | 13   | 21   | Jul 01 | 22   | 14   |  |  |
| Jun 16 | 64   | 30   | Jul 04 | 29   | 13   |  |  |

Figure 13. Ensemble of all the LWC profiles measured with dielectric instruments (light blue) from Apr 24, 2023 and Apr 04, 2024, i.e. the first dates for which significant mismatches between modelled and S1-acquired  $\sigma_0^{VV}$  values in 2023 (top row) and 2024 (bottom row), respectively. Melting calorimetry measurements (dark blue), including their associated uncertainty (dark blue shaded areas) as described in Barella et al. (2024), are shown for comparison. In 2024, a second simultaneous LWC profile using dielectric instruments (brown) was also carried out.

This uncertainty applies to the same cases as where "uncertainty in spatiotemporal LWC/TWC" applies, but it likely holds more weight when the TWC is slightly higher and the simulation variability according to LWC is lower (Apr 08-15, 2024; Apr 27 - May 05, 2024).

Another cause of significant discrepancy between recorded and modelled  $\sigma_0^{VV}$  in the presence of a mostly dry snowpack with a smooth surface may be the thawing of the soil. This process creates a thin layer of liquid water overlying the natural soil roughness or absorbed into the basal snow layer (Lombardo et al., 2025). The combination of snow wetness and roughness, as will be shown later in the paper, can be responsible for backscatter increases up to 7 dB. In Tab. 3 we refer to this kind of uncertainty as "wet soil scattering". This uncertainty potentially applies to the instances when the TWC is relatively low and the variability associated to LWC is high. Between Apr 04-27, 2024, our measurements show considerable amounts of liquid water at the soil interface with otherwise relatively dry snowpack and smooth surfaces (see Fig. 13). The lysimeter time series corroborates these measurements by detecting runoff start on Apr 08, 2024 (see Fig. 9e). However, we lack sufficient data in order to prove and explore this possible scattering source, therefore we only mention it as an hypothesis.







Two similar instances in 2023 (May 05-09) and 2024 (Apr 10 and 14) suggest another interesting phenomenon likely affecting simulation accuracy. On both these intervals, Fig. 12 shows very good agreement between recorded and modelled values of  $\sigma_0^{VV}$ , regardless of the chosen permittivity model and the variability associated to LWC. In both instances, surface roughness had just started developing on a wet snow surface (LWC>3%), with measured RMSH values between 3 and 4 mm (see Fig. 10b). Thereafter, spring snowfalls cover the early-stage roughness and the snow surface reverts to smooth with RMSH values between 1 and 2 mm (see Fig. 10a). On both years, the group of simulations following the spring snowfalls (i.e. May 11-15, 2023; May 21 and 22, 2024) show again strong biases when compared to S1 recordings. This bias is almost certainly due to the fact that the surface roughness which had started to develop was then buried below a smooth layer of new snow and it is not simulated by SMRT in the proposed configuration (see Sec 3.2). In Tab. 3 we labeled this phenomenon as "buried surface roughness".

Generally, simulations are in better agreement with S1 recordings when the measured surface RMSH is above 3 mm. Fig. 12 shows multiple groups of simulations where S1 retrievals from field data gain increasing accuracy with increasing RMSH on a wet surface, together with a decreasing dependence on the chosen permittivity model and the uncertainty associated to LWC (Apr 29 to May 09, 2023; May 19 to Jun 09, 2023; May 29 to Jul 01, 2024). These instances suggest that in conditions of increasing surface roughness on a wet snow surface, additional source of uncertainty in S1 retrievals from field data might be associated to the IEM (see Sec. 3.2) translating surface roughness in backscatter response and/or to point-wise panel measurements underestimating the surface roughness of the entire S1 cell. In Tab. 3 we labeled these sources as "uncertainty in IEM modelling" and "uncertainty in surface roughness measurements", respectively.

Interestingly, the S1 signal saturates at values of  $\sigma_0^{VV}$  of -22.4 and -23.7 dB for 2023 and 2024, respectively. These values are close to the nominal noise equivalent sigma naught (NESZ) of S1, i.e. -22 dB. The saturation of the signal is obtained by SMRT at much lower values, around  $\sim$ -30 dB, regardless of the chosen permittivity formulation.

#### 4.3.1 C-band radar backscatter sensitivity to the coupled evolution of surface roughness and liquid water content

To study the C-band radar backscatter sensitivity to the coupled evolution of surface roughness and LWC, we selected the date of Apr 16, 2024. On this date, we measured a melt event in the superficial 45 cm. The bottom part of the snowpack was homogeneously dry and was discretized as a one layer with the average of the scattering properties measured in the field. These

**Figure 14.** Sensitivity of the C-band radar backscatter to the coupled evolution of surface roughness (expressed by RMSH) and LWC. Panels (a, b, d, e) illustrate differences between two dielectric permittivity formulations – MEMLSv3 (a, b) and H-86 (c, d) – as well as the sensitivity to the local incidence angle (LIA) over cell 40.  $\sigma_0^{VV}$  responses are shown for 40° (solid lines) and 30° (dotted lines) incidence angles. Panels (c, f) show values of  $\left|\Delta\sigma_0^{VV}\right|_{40^\circ-30^\circ}$ , i.e., the differences between backscatter coefficients in (a, b) and (d, e), respectively. The real reference case is the snowpack layering observed on Apr 16, 2024: a melt event in the superficial 45 cm and an otherwise dry snowpack. The reported results are consecutive synthetic variations of LWC and roughness of the surface layer.

values are representative of a compacted snowpack structure at the beginning of the melt process: density of 428 kgm<sup>-3</sup>, SSA of 15.1 m<sup>2</sup>kg<sup>-1</sup> and temperature of -0.1°C. From this configuration, we prepared a series of synthetic snowpack variations with surface LWC increasing from 0 to 12%, and coupled each of them with a range of surface roughness RMSH increasing from 1 to 15 mm. These extremes represent a smooth surface typical of recent snowfall and the highly textured surface of fully formed suncups, respectively. To ensure consistency, we gradually increased the value of the second roughness parameter CL as well. To do so, we used an empirical logarithmic relationship extracted from field data between RMSH and CL, which we report in Fig. A1. However, this empirical relationship is based on a limited number of points (75 in total) which show larger spread for increasing values of RMSH. Therefore, we assume that the only two discontinuities in the experimental results (see Fig. 14a-c, RMSH=3 mm and LWC<sub>top</sub> =12%) can be explained considering this uncertainty. For clarity, these points were removed. All experiments were run with two incidence angles  $-30^{\circ}$  and  $40^{\circ}$  – which represent the overall range of angles between satellite



overpasses and the snow surface within the reference cell (see Fig. 2 and Tab. 1). The result of all the experiments is shown in Fig. 14, for both permittivity formulations.





In general, Fig. 14 shows that the intensity of the scattering response has a strong dependence on LWC for lower values of surface roughness (RMSH

**Figure A1.** Empirical logarithmic relationship fitted on field data between the surface roughness parameters of RMSH and CL, based on a total of N=75 values over the measurement campaigns of 2023 (yellow) and 2024 (light blue).