# Peer review of "Multitemporal analysis of Sentinel-1 backscatter during snow melt using high-resolution field measurements and radiative transfer modelling"

_EGUsphere, 2025_

## Author Comment (AC1)

**Revision to the manuscript preprint EGUSPHERE-2025-974**

The Authors would like to thank the Editor and the two anonymous Reviewers for the positive feedback to our study and for their detailed and constructive comments which will improve the quality of this manuscript. We are aware that reviews are a significant time investment and therefore especially appreciate their effort and feedback. The manuscript will undergo revision according to the Reviewers' comments. Please see below our responses, which are highlighted in blue.

**Response to the Reviewers**
* * *
**Reviewer 1**

**Reviewer Comment 1.1** — L376 – It was unclear to me how many layers were used for the SMRT simulations. From the text it seems layers were varied to account for the C-band wavelength, which is fine. However, a short statement outlining the range of layers or some statistics on this parameter would be good to see.

**Reply**: We thank the reviewer for pointing this out. We did not properly clarify this aspect in the text. We prepared Fig. 1, which we will insert into the Appendix. Fig. 1 represents the variability of the number of SMRT layers used for each simulation day as a function of the melting phase and the campaign year.

[Figure]

Figure 1: Variability of the number of SMRT layers used for each simulation day as a function of the melting phase and the campaign year.

At L379, we state: *To reduce the aforementioned sources of uncertainty, we chose to model the snowpack structure by stacking layers with a minimum thickness corresponding to the C-band wavelength, ensuring each layer had consistent average physical properties. These property-based layers*

*were identified automatically by means of a simple algorithm and then refined manually, with particular emphasis placed on LWC over the other variables.*

We will expand this section referring to Fig. 1 and explain the following concepts:

- The number of identified SMRT layers for each simulation day depends on the stage of the melting process. Fig. 1 shows that in dry snow conditions, the densely measured snow properties are often averaged into one single layer, given the absence of liquid water. As the snowpack starts moistening, the number of identified layers increases, as a function of the first formation of liquid water within the snowpack. Generally, the highest number of SMRT layers to model the snowpack is used during the ripening phase, as the LWC layering is at its most heterogeneous state during this phase, as a consequence of the progression of the wetting front. Later in the runoff stage, with the snowpack being fully saturated, the number of used SMRT layers decreases again, as a consequence of a more homogeneously moist snowpack.

- The number of identified SMRT layers also depends on the campaign year. Fig. 1 shows that the first campaign year has been modeled using ∼10% less layers than the second, on average. The presence of ice lenses helped to homogenize the distribution of liquid water within the snowpack, resulting in more uniformly wet layers near the surface and consistently drier sections toward the bottom. Without ice lenses, in 2024, the progression of liquid water into the snowpack was more heterogeneous, therefore requiring more modeling layers to remain as true as possible to the conditions observed in the field.

**Reviewer Comment 1.2** — Fig 2. - The colours in panel a) are not clear. The white area is skiable domain, however, has a greener shade to it. Recommend using a hash or some sort of pattern to denote this area.

Panel b) and Panel c) – I realize that if I zoom in the legend is more visible, however, it is difficult to see at 100% zoom. The legend is also identical for both panels. Therefore, one larger legend with font size increased would be an improvement.

**Reply**: We will improve the readability of manuscript's Fig. 2 according to the reviewer's suggestions.

**Reviewer Comment 1.3** — Figure 4. – This shows the setup for one SMRT simulation; however, nothing is marked as the SMRT layers used. Would this be possible to provide? Addresses my previous comment as well as providing some sort of example relate to the number of SMRT layers used.

**Reply**: We thank the reviewer for this remark. What we call "Physical layering" (horizontal yellow dashed lines) in manuscript's Fig. 4 are indeed the layers used in SMRT to simulate the radar backscattering on that specific day. We will modify the legend, the caption of the plot and the reference in the text in order to make this point more clear.

**Reviewer Comment 1.4** — Figure 5/6. – I think these are good figures illustrating the change in variables across the campaign. However, the text is not readable. Suggested changes: the different stages of snow melt could be included at the stop of only one pane as they are identical throughout the rest of the figure. Ideally the vertical lines could also be colour coded. Dates can be provided on only one pane as they are identical across. Also, font size is okay but a little small.

[Figure]

Figure 2: Modified version of manuscript's Fig. 8.

**Reply**: We thank the reviewer for these suggestions. We propose to leave the names of the melting phases at the top panel only and remove them from the rest of the panels, only leaving the separating dashed vertical lines. Our opinion is that by adding more color-coded lines, these plots might become overwhelming, therefore our preference is to leave the separating dashed vertical lines in gray. We agree that the dates on the x-axis of each panel might look redundant; however, in our opinion, they help improving the readability of the Figures, since we often mention different date ranges referred to different measured snow properties throughout the text. We will try to increase the font size.

**Reviewer Comment 1.5** — Figure 8 – Only one y-axis is needed between the figures. This would allow the size to be increased and increase readability.

**Reply**: We propose Fig. 2 as the new version of manuscript's Fig. 8, modified according to your suggestions. We merged the two panels into a single one, increasing the font size for better readability.

**Reviewer Comment 1.6** — Table 3 – Table layout is somewhat difficult to read. Horizontal outlines would aid in the interpretation of the data.

**Reply**: We will add horizontal outlines to improve the readability of the table.

**Reviewer Comment 1.7** — Figure 9 – This is a great figure. However, it is difficult to read. Two suggestions, 1) accompany the figure with a table which includes the specific explanations that way the table would be less busy, 2) use numbers or letters to point to specific events rather than having the arrows across the figure. This would again aid in improving the readability. Currently it takes too long to interpret it.

**Reply**: We agree with the reviewer that manuscript's Fig. 9 contains a great amount of information. It was rather hard to think about a way of conveying it while trying to avoid a chaotic result. We prepared a variation of this Figure according to your suggestions – see Fig. 3 and Tab. 1.

Table 1

| Group | Measured snow properties | | | Possible reasons for deviations |
| | TWC | LWC | RMSH | Possible scattering mechanisms |
|---|---|---|---|---|
| 1a | – | – | – | Backscattering increase due to soil thawing |
| 2a | <10 mm | <3% | ~1 mm | Uncertainty in spatiotemporal LWC/TWC
Scattering from surface structures due to melt-refreeze
Underestimation of surface roughness
Scattering from the wet soil |
| 3a | >10 mm | >3% | 1 → 4 mm | Uncertainty in spatiotemporal LWC
Uncertainties in roughness measurements
Uncertainties in IEM modeling |
| 4a | >10 mm | >3% | 3~4 mm | Uncertainty in spatiotemporal LWC
Uncertainties in roughness measurements
Uncertainties in IEM modeling |
| 5a | >10 mm | >3% | ~1 mm | New snowfall on a rough surface: "buried roughness" |
| 6a | <10 mm | <3% | ~1 mm | Cold spell
Uncertainty in spatiotemporal LWC
Scattering from surface structures due to melt-refreeze
Underestimation of surface roughness
Scattering from wet soil |
| 7a | >10 mm | >3% | >4 mm | Uncertainty in spatiotemporal LWC
Uncertainties in roughness measurements
Uncertainties in IEM modeling |
| 1b | – | – | – | Backscattering increase due to soil thawing |
| 2b | <10 mm | <3% | ~1 mm | Uncertainty in spatiotemporal LWC/TWC
Scattering from surface structures due to melt-refreeze
Scattering from the wet soil |
| 3b | >10 mm | >3% | ~1 mm | Uncertainty in spatiotemporal LWC
Scattering from surface structures due to melt-refreeze
Underestimation of surface roughness |
| 4b | Varying, <10 mm | <3% | ~1 mm | Cold spell
Uncertainty in spatiotemporal LWC
Scattering from surface structures due to melt-refreeze
Underestimation of surface roughness
Scattering from wet soil |
| 5b | >10 mm | >3% | ~1 mm | Uncertainty in spatiotemporal LWC
Scattering from surface structures due to melt-refreeze
Underestimation of surface roughness |
| 6b | >10 mm | >3% | ~3 mm | Uncertainty in spatiotemporal LWC
Uncertainties in roughness measurements
Uncertainties in IEM modeling |
| 7b | >10 mm | >3% | ~1 mm | New snowfall on a rough surface: "buried roughness" |
| 8b | >10 mm | >3% | >4 mm | Uncertainty in spatiotemporal LWC
Uncertainties in roughness measurements
Uncertainties in IEM modeling |

[Figure]

Figure 3: Modified version of manuscript's Fig. 9.

Our opinion, however, is that by off-loading the descriptions to a table the readability doesn't necessarily improve. In the old version, all information is on the same figure – albeit a figure which takes time to interpret. In the new version, the figure is indeed lighter, but the reader has to toggle between the figure and the table, henceforth the interpretation of the information is not necessarily faster or easier.

**Reviewer Comment 1.8** — Table 4 – Similar comment to table 3.

**Reply**: We will add horizontal outlines to improve the readability of the table.

**Reviewer Comment 1.9** — Figure 11 – The bottom pane on this figure is unnecessary – it does not add anything to the interpretation. Simply providing the LWC for the top layer would be sufficient. Also, the LWC legend is unnecessary on the bottom. The RMSH legend is good.

**Reply**: We will modify manuscript's Fig. 11 according to the reviewer's suggestions.

**Reviewer 2**

**Reviewer Comment 2.1** — The radiometric accuracy for the backscatter data of the reference cell needs to be specified in detail. The reference to Marin et al. (2020) does not provide specific information on radiometric uncertainty for this particular cell of 20 m x 20 m extent. In S1 IW

mode data it covers only about 4 independent samples (looks), resulting in high speckle-related uncertainty. Marin et al. apply for speckle reduction a multitemporal filter of 11 x 11 pixels and a gamma-MAP filter of 3 x 3 pixels. In both cases the radiometry is preserved if the window covers a homogeneous distributed target. This is not the case in the area surrounding of the refence cell. For the multitemporal filter different temporal response within the window may introduce additional radiometric biases for sigma-0 of individual pixels and dates.

**Reply**:

We appreciate the reviewer's insightful comment regarding the characterization of radiometric uncertainty, which allows us to provide important clarification. We agree that thoroughly addressing radiometric uncertainty is crucial for our study, especially concerning the impact of speckle.

Indeed, our signal analysis and subsequent comparison with model results are influenced by several sources of uncertainty. As the reviewer rightly points out, the radar instrument itself is a significant contributor. ESA calibration campaigns indicate a nominal uncertainty within a $3\sigma$ of $1.0$ dB [1–3]. However, additional pre-processing steps also contribute to the overall uncertainty. These include not only the application of despeckle filters but also crucial operations like terrain correction and radiometric normalization, which are particularly challenging in mountainous regions due to complex topography. Other steps such as thermal noise removal (important in conditions of high absorption like wet snow) and spatial interpolations further add to this complexity. Consequently, a detailed specification of the overall radiometric accuracy becomes quite complex and is beyond the scope of this paper, which we have earmarked for future dedicated work.

Nonetheless, we believe it is valuable to illustrate the temporal behavior of backscattering over the target cell and demonstrate the importance of speckle denoising. For this study, we opted for the multi-temporal filter proposed by Quegan and Yu in [4] to mitigate speckle impact. The Quegan and Yu filter offers a powerful yet simple approach for denoising multi-temporal stacks. Its original implementation involves local averages of intensity values for each date (akin to local spatial multi-looking), this can lead to a loss of resolution, blurring strong targets and edges, and ultimately impacting the global multi-temporal result as pointed out by the reviewer. Indeed, we acknowledge that more complex and robust despeckling methods are now available (e.g., [5]) and can be considered for future work. However, when the target is presumed to be time-invariant, such as during the winter period (with stable scatterer positions, snow cover, temperatures well below 0°C, and frozen soil ensuring no soil moisture variation), the pixels in our study exhibited stable behavior. This stability was also observed during dry periods in summer. In these specific cases, the standard deviation was within $1.0$ dB, which aligns well with S1's nominal radiometric uncertainty.

On the other hand, Figure 4 illustrates the behavior for track 168 during the melting period when differences in time are present due to wetting of the snow. As evident, the differences between various window sizes and the unfiltered signals are minimal, justifying our pragmatic choice of an 11 x 11 pixel window: this choice aims to reduce uncertainty due to speckle while (trying to) introducing minimal bias. Crucially, we want to clarify that the gamma-MAP filter (as used in Marin et al. 2020) was not applied in this study – we will make this distinction clearer in the revised version of the manuscript.

Finally, we acknowledge that the primary source of radiometric uncertainty, particularly during the melting period, originates from the presence of liquid water content (LWC). Estimating radiometric uncertainty under such conditions is extremely difficult, given LWC high heterogeneity across the entire resolution cell, as highlighted already in the manuscript. Therefore, a rigorous uncertainty estimation without a precise reference for LWC is inherently challenging and falls outside the scope of this paper. We will introduce the discussion of radiometric uncertainty more thoroughly in the revised manuscript

and clearly outline this as a limitation of our current study, highlighting it as an important direction for future dedicated research.

[Figure]

Figure 4: Effect of the multitemporal filter, with different window sizes, to the backscatering signal for track 168 during the melting season of 2023-2024.

**Reviewer Comment 2.2** — The representativeness of the S1 signatures of the reference cell in respect to the area in its surroundings should be assessed by analysing also data of other cells, preferably covering some larger area. Considering Fig. 2, it is unclear why cell 39 is used as single reference case. The data of cell 39 do not show a distinct incidence angle dependence and track 117 shows a different behaviour in the two years. The data of cell 18 of the two years, for example, are consistent. According to the elevation contour lines, this cell is located on an east-facing slope, and the two tracks of the ascending orbit show lower sigma-0 values, as expected for a back-slope. This may offers a possibility for studying impacts of the incidence angle.

**Reply**: We thank the reviewer for their comments on the representativeness of the selected S1 cell. These remarks helped us to corroborate the premises of our analyses.

In the manuscript (Sec. 4.2), we tried to highlight the heterogeneity of LWC measurements (even when measuring simultaneously). Such heterogeneity has been previously analyzed and consolidated in both laboratory settings and field studies [6, 7]. It is true that the representativeness of the chosen S1 reference cell would benefit from the comparisons to other neighboring cells. However, the range of locations within the field site being undisturbed, potentially matching the snow characteristics measured in the snow pit and not affected by double-bounce effects from big metallic structures nearby is rather small and likely limited to cells 32, 38, 39, 40. In the reviewed version of the manuscript, we will show a comparison between the S1 signatures of the selected reference cell with respect to these neighboring cells.

As explained in Section 3.1 of the manuscript, cell 39 was initially chosen primarily because it is closest to our measurement field and therefore most likely to represent the snow conditions accurately. However, as the reviewer rightfully points out, the incidence angle generally plays an important role in

wet snow conditions. Therefore, in the reviewed version of the manuscript, we will change the reference cell to cell 40. Cell 40 covers a flat area right next to our measurement field, and the snow surface there is totally undisturbed because of the presence of a terrestrial laser scanner taking point cloud acquisitions of the snow surface at high temporal resolution. Therefore, we could hypothesize that the snow properties are as similar as possible as those we have measured. Besides, the data of cell 40 also show relatively distinct incidence angle dependence as for cell 18. We have three major concerns with the choice of cell 18 as a reference cell. In the first place, although it lies officially outside of the skiable domain, it is not uncommon for skiers to traverse that area, especially for employees on duty for daily snow measurements throughout the snow season. This highlights the need for a more precise delineation of the skiable domain in Fig. 2 of the manuscript, which we will address in the revised version. On the other hand, cell 18 belongs to a east-faced slope ($\sim$11°), where the snow cover becomes patchy and disappears earlier with respect to the flatter area ($\sim$5.5°) where the preferred cells are situated (see Fig. 6). This could introduce ambiguities in our discussion of backscattering increases related to the development of surface roughness. Finally, cell 18 lies $\sim$90 meters away from the area where we performed our snow pits. To our knowledge, the greatest horizontal distance over which LWC variability has been characterized is 5 meters on relatively flat terrain [6]. Therefore, selecting a greater horizontal distance and slope difference than what we used could potentially introduce additional uncertainties in our analysis that we would be unable to quantify.

The impact of incidence angles was not explicitly designed as a part of this study, primarily because this topic has already been extensively investigated in the literature already cited in the manuscript – and further complemented by the reviewer's suggestions [8–10]. These studies primarily rely on tower-based radiometers and radars. Tower-based settings offer a level of precision and control that is not feasible for radars installed on satellite platforms overlooking 20 x 20 meter cells. In our case, the region matching the characteristics of the snow pits' area is small and primarily flat, therefore the range of local incidence angles suitable for a comprehensive study is relatively constrained (see Fig. 5), thus limiting the possibility to investigate its impact.

Moreover, due to the significant heterogeneity of LWC, a comprehensive study covering a sufficiently wide range of local incidence angles would ideally require a separate ground reference for each cell. However, this was not feasible given the time and resource demands already involved in conducting the analysis at a single location. The use of different tracks was done primarily in order to maximize the number of possible comparisons under different snow conditions, also given the failure of Sentinel-1B. As explained in Sec 3.2 of the manuscript, the incidence angle is not responsible for deviations between backscattering simulations and S1 recordings. This is because the SMRT model is provided with either the exact incidence angle of S1 (when measurements and satellite overpasses coincide) or the incidence angle of the temporally closest overpass (when they do not). However, as the reviewer correctly notes in the following Comment 2.3, the incidence angle is relevant when analyzing roughness effects on wet snow surfaces, as we do in Section 4.3.1 of the manuscript by means of synthetic experiments. In the reviewed version, we will improve this section by repeating such experiments across the range of incidence angles of cell 40. The results will be discussed in the context of the prior studies that the reviewer mentioned [8,9] – although the surface roughness was not measured in these studies, but only assessed.

We will ensure that each of the above points is thoroughly addressed and that our choices are clearly justified in the revised version of the manuscript.

**Reviewer Comment 2.3** — For relating the backscatter modelling results to specific melt

phases and for interpretation of the observed backscatter signatures, first of all it is important providing information on the backscatter contributions of the individual snow layers in dependence of the liquid water content. The relative contributions to total backscatter in dependence of depth below the snow surface should be specified for typical snow profiles shown in Figs. 5 and 6. Another concern is the limited information regarding properties of the scattering elements in the individual layers and the parametrization of the dense medium effect. In particular for the early melt phases, the properties of the top snow layer can be quite different, depending on size and shape of water inclusions (Arslan et al., 2003). Furthermore, information on the impact of incidence angle dependence of backscatter would be of interest, being of relevance for assessing roughness effects of wet snow surfaces.

**Reply**: We thank the reviewer for these valuable comments, which highlight issues that were not sufficiently addressed in the initial version of the manuscript.

In a previous version of our work, we included a section addressing the relative contribution to the total backscattering of individual snow layers, as a function of the measured liquid water content. We finally decided to remove it, because the solidity of such analysis is strongly dependent on two main aspects. One of them is the lack of a consolidated wet snow permittivity formulation, which as explained in Section 3.2 of the manuscript, remains an unresolved issue in the field of electromagnetic modeling of wet snow surfaces. In the course of our analysis, we compare two formulations which were previously validated against real-world C-band data [11–13]. However, these two formulations diverge significantly at C-band, especially for what concerns the prediction of the imaginary part of the permittivity, which governs absorption losses. Such divergence becomes more pronounced for increasing values of LWC, as we show in Fig. 3 of the manuscript. On the other hand, we were unable to measure and model larger (superficial or internal) snow structures potentially causing additional scattering effects, such as crusts, which might have an important effect on the total recorded backscattering at the cell-scale. In the reviewed version of the manuscript, we will make sure to highlight this issue more clearly.

Furthermore, we agree with the reviewer that our dataset lacks information about the shape of the water inclusions per layer, and we thank them for providing this reference, which we overlooked in our literature review. To our knowledge, neutron radiography can be used to distinguish water from ice, however, it requires either prior knowledge of the dry density or other measurements such as energy-selective neutron radiography [14]. In such experiments, the initial conditions need to be carefully controlled, therefore requiring a laboratory setting. [15] recently presented the nuclear MRI rapid profiling technique, which allowed measuring different states of wetting snow and therefore the shape of the water inclusions at unprecedented resolutions. This technique also depends on a controlled laboratory environment. At the time our measurement campaign was designed and conducted, these methods did not yet exist – let alone their applicability in the field, which is still entirely unknown. However, this would be an extremely promising development for future research and similar studies. In the revised manuscript, we will acknowledge the additional source of uncertainty arising from the practical infeasibility of characterizing the shape of water inclusions in an experimental field study, and suggest exploring the feasibility of methodologies as [15] in the field as potential future developments.

Regarding the incidence angle effects, we refer to our previous Reply 2.2.

**Reviewer Comment 2.4** — Introduction and Discussion: During three winters Strozzi and Mätzler (1997; 1998) performed at the same test field above Davos C- and Ka-band backscatter measurements. Reference 1 (1997) is briefly cited in the manuscript, reference 2 (1998) is not cited. Results of these measurements are relevant within the context of the work presented by Carletti et

al. and key points should be mentioned. Among issues addressed in the two papers are impacts of surface roughness and refrozen snow crusts based on measurements at different incidence angles, and the response to liquid water content.

**Reply**: We thank the reviewer for this comment – we have missed the above mentioned article in our literature review. We will make sure to reference it in the Introduction and Discussion of the revised version of the manuscript.

**Reviewer Comment 2.5** — P4, L121: The sensitivity to snow wetness depends on the local incidence angle, not directly on slope steepness. On steep fore-slopes the sensitivity is low.

**Reply**: We thank the reviewer for this comment. We will correct this sentence.

**Reviewer Comment 2.6** — P8, L212: A cutter of 55 cm length may not be suitable for resolving the density differences between individual layers that may be quite thin during the different melt phases. Please explain the limits in vertical resolution.

**Reply**: The cylinder cutter was only used for manual measurements of bulk snow water equivalent, which served as validation for the automatic sensor. As explained in Section 2.1.2 of the manuscript, manual density measurements were performed using a box cutter with a vertical dimension of 3 cm. This method allowed us to precisely measure density differences between individual snow layers (see Fig. 4 of the manuscript). The resulting density profiles were then used as part of the inputs to the radiative transfer model to simulate the S1 backscattering.

**Reviewer Comment 2.7** — P9, Table 1: The elevation contour lines of Fig. 2a indicate different slope steepness within the test field. Please explain to which points the cited incidence angles refer and show the overall range of angles for the test field.

**Reply**: We thank the reviewer for noting that in the original text there was ambiguity between "test field" and the reference cell actually chosen for the analysis. We will clarify that the incidence angles reported in Tab. 1 of the manuscript refer to the selected reference cell. Moreover, to address the reviewer's concerns about the influence of the incidence angle in Comment 2.2, we prepared Fig. 5, which we will include in the Appendix.

**Reviewer Comment 2.8** — P11, Fig: 2: Incidence angles of cells on sloping show lower sigma-0 for ascending orbits, as to be expected for back-slopes. Consequently, the difference in sigma-0 between individual tracks may offer the possibility for exploring incidence angle effects.

**Reply**: We thank the reviewer for this comment. In Reply 2.2, we have explained why studying the effects of the incidence angle is quite impractical in a study designed like ours, primarily due to the extreme heterogeneity of the LWC. However, we will propose this topic as one of the future developments of our work.

**Reviewer Comment 2.9** — P12, L328: The characterization of snow microstructure is a critical issue for snow backscatter modelling. Exponential correlation functions have major deficiencies, in particular for multi-size and sticky cases (Chang et al., 2016). Furthermore, the phase functions of snow with liquid water inclusions are quite different from that of dry snow (Arslan et al., 2003).

**Reply**: We thank the reviewer for their comments about the characterization of snow microstructure, and for providing important literature reference which we overlooked.

The choice of an exponential correlation function is explained at P12, L325-330 of the manuscript. In these lines, we refer to the study of [16], specifically to Section 2.2, where the authors illustrate the unifying role of the microwave grain size ($\ell_{MW}$) at low frequencies such as the C-band. In detail, the authors explain how $\ell_{MW}$ can be computed for different analytical forms of various auto-covariance functions (i.e. the exponential and the sticky hard spheres) and then related to specific parameters of such forms. Notably, it is not guaranteed that such analytical forms always exist, but they do for the aforementioned formulations. In such cases, the analytical expressions of $\ell_{MW}$ make the different representations of microstructure comparable. Most importantly, it is guaranteed that different microstructure representations predict the same scattering amplitude in the low frequency limit when the same value of $\ell_{MW}$ is used as input. Therefore, according to these findings, the choice of the best representation of snow microstructure becomes a secondary problem with respect to measuring $\ell_{MW}$ in order to predict snow scattering in the microwave domain. $\ell_{MW}$ is computed as the product of the Porod length $\ell_P$ and the polydispersity $k$. As explained at P12, L325-330, we computed $\ell_P$ from our detailed field measurements and chose an empirical value of $k$ according to the findings of [16], which are based on a comprehensive set of $\mu$-CT scans covering a wide variety of Alpine snow samples with convex grains, among which rounded grains and melt forms.

The reviewer correctly points out that phase functions of wet snow differ from those of dry snow. We will address this source of uncertainty in the revised version of the manuscript.

**Reviewer Comment 2.10** — P14, L363: In particular during a main part of winter 2023-2024 the base of the snowpack shows zero deg. temperature, implying unfrozen ground. This goes on throughout the snowmelt periods.

**Reply**: We thank the reviewer for this remark. The sentence they refer to is actually formulated poorly. Indeed, we do not model the soil as a frozen surface. Using the functions available in SMRT, we model the substrate as a reflecting surface with a given value of backscattering. In dry snow conditions, on days when manual measurements and satellite overlooks coincide, we assign the S1 recorded backscattering value to the substrate, assuming that dry snow is transparent to radar waves at C-band and that therefore the soil is the only contribution to the total backscattering. In wet snow conditions (or in dry snow conditions, when there is no concomitance between measurements and satellite overlooks), we assign a fixed value of backscattering to our substrate, which we compute as the average value in dry snow conditions of each individual track (incidence angle). Notably, SMRT offers the possibility to compute the backscattering from the soil, however, it requires a series of detailed information that are spatially heterogeneous and would have been nearly impossible to retrieve continuously over the course of our campaign. These properties include the soil moisture, the relative sand content, the relative clay content, the soil content in dry matter, and other geometrical parameters such as the roughness and the correlation length.

**Reviewer Comment 2.11** — P16 Fig. 5: Between mid-April and early June 2023 sigma-0 of the two ascending tracks shows consistent differences of 3 to 5 dB. Please explain the reason. The high sigma-0 values are probably from track 117 which shows high variance in 2023-2024, differing from 2022-2023 (Fig. 2b).

**Reply**: We thank the reviewer for this remark, which we will address in the revised version of the manuscript.

**Reviewer Comment 2.12** — P20, Table 2: Please specify the incidence angle. Besides, the validity of these numbers in respect to other incidence angles would be of interest.

**Reply**: We thank the reviewer for this comment. The values in manuscript's Tab. 2 were averaged over the 4 incidence angles. We enhanced this analysis and highlighted the dependence on the incidence angle.

**Reviewer Comment 2.13** — P 27, Fig. 11: Please specify the incidence angle. Strozzi and Mätzler (1997) show the incidence angle dependence of backscatter of wet snow (for smooth and rough surfaces) based on backscatter measurements at the same test site.

**Reply**: The used incidence angle is specified in the text, however, according to your suggestion, we will improve our analysis by repeating the simulations for the incidence angles overlooking cell 40. While comparing the various datasets is of great interest, the considerable LWC heterogeneity complicates direct comparisons. Moreover, our dataset includes time series of measured surface roughness, whereas [8, 9] provided either visual estimates or just estimates.

**Reviewer Comment 2.14** — P29, L625: The development of surface roughness after start of the snowmelt period depends also on the sequence and intensity of snowfall events, varying from year to year.

**Reply**: We thank the reviewer for this remark, we will rephrase the sentence.

**Reviewer Comment 2.15** — P29, L641ff: The parametrization of the scattering elements may as well be a reason for differences between recorded and modelled backscatter (see e.g. Arslan et al., 2003; Chang et al., 2016).

**Reply**: As discussed above, we will make sure to mention this limitation in the Discussion.

**References**

[1] R. Torres, P. Snoeij, D. Geudtner, D. Bibby, M. Davidson, E. Attema, P. Potin, B. Rommen, N. Floury, M. Brown, I. N. Traver, P. Deghaye, B. Duesmann, B. Rosich, N. Miranda, C. Bruno, M. L'Abbate, R. Croci, A. Pietropaolo, M. Huchler, and F. Rostan, "Gmes sentinel-1 mission," *Remote Sensing of Environment*, vol. 120, pp. 9–24, 2012, the Sentinel Missions - New Opportunities for Science. [Online]. Available: https://www.sciencedirect.com/science/article/pii/S0034425712000600

[2] M. Schwerdt, K. Schmidt, N. Tous Ramon, P. Klenk, N. Yague-Martinez, P. Prats-Iraola, M. Zink, and D. Geudtner, "Independent system calibration of sentinel-1b," *Remote Sensing*, vol. 9, no. 6, 2017.

[3] H.-J. F. Benninga, R. van der Velde, and Z. Su, "Sentinel-1 soil moisture content and its uncertainty over sparsely vegetated fields," *Journal of Hydrology X*, vol. 9, p. 100066, 2020. [Online]. Available: https://www.sciencedirect.com/science/article/pii/S2589915520300171

[4] S. Quegan and J. J. Yu, "Filtering of multichannel sar images," *IEEE Transactions on Geoscience and Remote Sensing*, vol. 39, no. 11, pp. 2373–2379, 2001.

[5] I. Meraoumia, E. Dalsasso, L. Denis, R. Abergel, and F. Tupin, "Multitemporal speckle reduction with self-supervised deep neural networks," *IEEE Transactions on Geoscience and Remote Sensing*, vol. 61, pp. 1–14, 2023.

[6] F. Techel and C. Pielmeier, "Point observations of liquid water content in wet snow ndash; investigating methodical, spatial and temporal aspects," *The Cryosphere*, vol. 5, no. 2, pp. 405–418, 2011. [Online]. Available: https://tc.copernicus.org/articles/5/405/2011/

[7] F. Avanzi, H. Hirashima, S. Yamaguchi, T. Katsushima, and C. De Michele, "Observations of capillary barriers and preferential flow in layered snow during cold laboratory experiments," *The Cryosphere*, vol. 10, no. 5, pp. 2013–2026, 2016. [Online]. Available: https://tc.copernicus.org/articles/10/2013/2016/

[8] T. Strozzi, A. Wiesmann, and C. Mätzler, "Active microwave signatures of snow covers at 5.3 and 35 ghz," *Radio Science*, vol. 32, no. 2, pp. 479–495, 1997.

[9] T. Strozzi and C. Mätzler, "Backscattering measurements of alpine snowcovers at 5.3 and 35 ghz," *IEEE Trans. Geosci. Remote. Sens.*, vol. 36, pp. 838–848, 1998. [Online]. Available: https://api.semanticscholar.org/CorpusID:10925190

[10] A. Arslan, H. Wang, J. Pulliainen, and M. Hallikainen, "Scattering from wet snow by applying strong fluctuation theory," *Journal of Electromagnetic Waves and Applications*, vol. 17, no. 7, pp. 1009–1024, 2003.

[11] M. Hallikainen, F. Ulaby, and M. Abdelrazik, "Dielectric properties of snow in the 3 to 37 ghz range," *IEEE Transactions on Antennas and Propagation*, vol. 34, no. 11, pp. 1329–1340, 1986.

[12] T. Achammer and A. Denoth, "Snow dielectric properties: from dc to microwave x-band," *Annals of Glaciology*, vol. 19, p. 92–96, 1994.

[13] J. Kendra, K. Sarabandi, and F. Ulaby, "Radar measurements of snow: Experiment and analysis," *Geoscience and Remote Sensing, IEEE Transactions on*, vol. 36, pp. 864 – 879, 06 1998.

[14] M. Lombardo, P. Lehmann, A. Kaestner, A. Fees, A. Van Herwijnen, and J. Schweizer, "A method for imaging water transport in soil–snow systems with neutron radiography," *Annals of Glaciology*, vol. 65, p. e8, 2025.

[15] Q. Krol, E. Scherrer, M. Skuntz, S. Codd, A. Hansen, and J. Seymour, "Rapid mri profiling of liquid water content in snow: Melt and stability during first wetting and rain on snow events," in *Proceedings of the International Snow Science Workshop*. Tromsø, Norway: Montana State University Library, 2024. [Online]. Available: https://arc.lib.montana.edu/snow-science/objects/ISSW2024_O3.11.pdf

[16] G. Picard, H. Löwe, F. Domine, L. Arnaud, F. Larue, V. Favier, E. Le Meur, E. Lefebvre, J. Savarino, and A. Royer, "The microwave snow grain size: A new concept to predict satellite observations over snow-covered regions," *AGU Advances*, vol. 3, no. 4, p. e2021AV000630, 2022, e2021AV000630 2021AV000630. [Online]. Available: https://agupubs.onlinelibrary.wiley.com/doi/abs/10.1029/2021AV000630

[Figure]

Figure 5: Range of local incidence angles over the Weissfluhjoch measurement station for each S1 afternoon (A) and morning (M) track.

[Figure]

Figure 6: Webcam acquisition of the field site of Weissfluhjoch (June 11th, 2025 at 16:00:00). Area 1 (in blue) indicates the approximate location of cell 18. During snow ablation, this section shows earlier snow disappearance with respect to the flatter locations where cells 32, 38, 39, 40 belong, highlighted by area 2 (in red).

---

## Author Response (AR1)

**Revision to the manuscript preprint EGUSPHERE-2025-974**

The Authors would like to thank the Editor and the two anonymous Reviewers for the positive feedback to our study and for their detailed and constructive comments which significantly improved the quality of this manuscript. We are aware that reviews are a significant time investment and therefore especially appreciate their effort and feedback. The manuscript was revised according to the Reviewers' comments. Please see below our responses, which are highlighted in blue, with significant additions being highlighted in green italics. To avoid confusion with the manuscript's labelling, the references to the Figures and Tables in the present document are reported in bold.

**Response to the Reviewers**

**Reviewer 1**

Reviewer Comment 1.1 — L376 – It was unclear to me how many layers were used for the SMRT simulations. From the text it seems layers were varied to account for the C-band wavelength, which is fine. However, a short statement outlining the range of layers or some statistics on this parameter would be good to see.

**Reply**: We thank the reviewer for this comment. We added **Fig. 1** to the manuscript, to represent the variability of the number of the modeling layers used in SMRT for each simulation day as a function of the melting phase and campaign year.

Figure 1: Variability of the number of modeling layers in SMRT used for each simulation day as a function of the melting phase and the campaign year.

In the revised version of the manuscript (L487-499), we added the following explanatory paragraph.

(...) Fig. 7 (Fig. 1) shows that the number of layers used for each SMRT simulation varied between 1 and 14, with a marked dependence on the stage of the melting process and on the campaign year. In dry snow conditions, the densely measured snow properties are practically always averaged into one single layer, given the

absence of liquid water. As the snowpack starts moistening, the number of distinct layers increases, as a function of the first formation of liquid water within the snowpack. The highest number of layers required in SMRT to model the snowpack is used during the ripening phase, as the LWC layering is at its most heterogeneous state during this phase, as a consequence of the progression of the wetting front. Later in the runoff stage, with the snowpack being fully saturated, the number of used SMRT layers decreases again, as a consequence of a more homogeneously moist snowpack. On the other hand, Fig. 7 (Fig. 1) shows that during the ripening phase, the first campaign year has been modeled using  $\sim 30\%$  less layers than the second, on average. The presence of ice lenses helped to homogenize the distribution of liquid water within the snowpack, resulting in more uniformly wet layers near the surface and consistently drier sections toward the bottom. Without ice lenses, in 2024, the progression of liquid water into the snowpack was more heterogeneous, therefore requiring more layers in the model to remain as true as possible to the conditions observed in the field. (...)

**Reviewer Comment 1.2** — Fig 2. - The colours in panel a) are not clear. The white area is skiable domain, however, has a greener shade to it. Recommend using a hash or some sort of pattern to denote this area.

Panel b) and Panel c) – I realize that if I zoom in the legend is more visible, however, it is difficult to see at 100% zoom. The legend is also identical for both panels. Therefore, one larger legend with font size increased would be an improvement.

**Reply**: We thank the reviewer for this comment. We removed the skiable domain from ex-Fig. 2 as it was not so relevant for the discussion. Ex panels (b, c) are now panels (a, b) of **Fig. 2**. We improved the readability of the legend.

**Reviewer Comment 1.3** — Figure 4. – This shows the setup for one SMRT simulation; however, nothing is marked as the SMRT layers used. Would this be possible to provide? Addresses my previous comment as well as providing some sort of example relate to the number of SMRT layers used.

**Reply**: We thank the reviewer for this remark. We made a new version of ex-Fig. 4 (see **Fig. 3**, where the modeling layers in SMRT are defined clearly.)

Reviewer Comment 1.4 — Figure 5/6. —I think these are good figures illustrating the change in variables across the campaign. However, the text is not readable. Suggested changes: the different stages of snow melt could be included at the stop of only one pane as they are identical throughout the rest of the figure. Ideally the vertical lines could also be colour coded. Dates can be provided on only one pane as they are identical across. Also, font size is okay but a little small.

**Reply**: We thank the reviewer for this suggestion. In the new version of such Figures (now Fig. 8 and 9), we left the names of the melting phases at the top panel only and removed them from the other panels, leaving the separating dashed vertical lines. Our opinion is that by adding more color-coded lines, these plots would become overwhelming, therefore our preference was to leave the separating dashed vertical lines in gray. We agree that the dates on the x-axis of each panel look redundant; however, in our opinion, they help improving the readability of the Figures, since we often mention different date ranges referring to different measured snow properties throughout the text. We increased the font size.

**Reviewer Comment 1.5** — Figure 8 – Only one y-axis is needed between the figures. This would allow the size to be increased and increase readability.

**Reply**: We thank the reviewer for this comment. We made a new version of ex-Fig. 8 (see **Fig. 4**), where we implemented these suggestions to improve readability.

**Reviewer Comment 1.6** — Table 3 – Table layout is somewhat difficult to read. Horizontal outlines would aid in the interpretation of the data.

Reply: We thank the reviewer for this suggestion. We improved the readability of all tables by adding gridlines.

Reviewer Comment 1.7 — Figure 9 – This is a great figure. However, it is difficult to read. Two suggestions, 1) accompany the figure with a table which includes the specific explanations that way the table would be less busy, 2) use numbers or letters to point to specific events rather than having the arrows across the figure. This would again aid in improving the readability. Currently it takes too long to interpret it.

**Reply**: We thank the reviewer for these important suggestions which helped us improving the readability of a key figure for this paper. We agree that ex-Fig. 9 was too busy and therefore difficult to read and interpret. We prepared a new version of it (see **Fig. 5**) where groups are labeled with numbers and letters. **Fig 5** is now accompanied by a table (see **Tab. 1**) where, for each label, scattering mechanisms and/or deviations between S1 recordings and model results are explained.

Table 1: Supplementary information to Fig. 12 (**Fig. 5**): measured values of TWC, LWC, RMSH, noteworthy events for scattering (such as cold spells or late snowfalls), and explanations to the mismatch between modeled and recorded S1 backscatter signatures.

| Group | TWC                 | LWC | RMSH                  | Event                                                              | Source(s) of inconsistency, scattering mechanism                                                                                                                                                          |  |  |
|-------|---------------------|-----|-----------------------|--------------------------------------------------------------------|-----------------------------------------------------------------------------------------------------------------------------------------------------------------------------------------------------------|--|--|
| 1a    | _                   |     | _                     | Soil thawing                                                       | - Backscattering increase due to soil thawing                                                                                                                                                             |  |  |
| 2a    | <10 mm              | <3% | 1 mm                  | Snowpack moistening
Smooth surface                              |  <li>Uncertainty in spatiotemporal LWC/TWC</li> <li>Scattering from surface structures (melt-refreeze)</li> <li>Surface roughness underestimation</li> <li>Wet soil scattering</li>              |  |  |
| 3a    | >10 mm              | >3% | 1→4 mm                | Snowpack ripening
Formation of surface roughness                |  <li>Uncertainty in spatiotemporal LWC/TWC</li> <li>Uncertainty in surface roughness measurements</li> <li>Uncertainty in IEM modeling</li>                                                      |  |  |
| 4a    | >10 mm              | >3% | 3~4 mm                | Snowpack ripening
Increasing surface roughness                  |  <li>Uncertainty in spatiotemporal LWC/TWC</li> <li>Uncertainty in surface roughness measurements</li> <li>Uncertainty in IEM modeling</li>                                                      |  |  |
| 5a    | >10 mm              | >3% | ${\sim}1~\mathrm{mm}$ | New snowfall on a wet snowpack
Well-developed surface roughness | - "Buried surface roughness"                                                                                                                                                                              |  |  |
| 6a    | <10 mm              | <3% | $\sim 1 \text{ mm}$   | Cold spell (partial snowpack refreeze)
Smooth surface           |  <li>Uncertainty in spatiotemporal LWC/TWC</li> <li>Scattering from surface structures (melt-refreeze)</li> <li>Uncertainty in surface roughness measurements</li> <li>Wet soil scattering</li>  |  |  |
| 7a    | >10 mm              | >3% | >4 mm                 | Wet snowpack
Fully-formed suncups                               |  <li>Uncertainty in spatiotemporal LWC/TWC</li> <li>Uncertainty in surface roughness measurements</li> <li>Uncertainty in IEM modeling</li>                                                      |  |  |
| 1b    | <10 mm              | <3% | ~1 mm                 | Snowpack moistening
Smooth surface                              |  <li>Uncertainty in spatiotemporal LWC/TWC</li> <li>Scattering from surface structures (melt-refreeze)</li> <li>Surface roughness underestimation</li> <li>Wet soil scattering</li>              |  |  |
| 2b    | >10 mm              | >3% | ~1 mm                 | Snowpack moistening
Smooth surface                              |  <li>Uncertainty in spatiotemporal LWC/TWC</li> <li>Scattering from surface structures (melt-refreeze)</li> <li>Surface roughness underestimation</li>                                           |  |  |
| 3b    | <10 mm
(Varying) | <3% | ~1 mm                 | Cold spell (partial snowpack refreeze) Smooth surface              |  <li>Uncertainty in spatiotemporal LWC/TWC</li> <li>Scattering from surface structures (melt-refreeze)</li> <li>Surface roughness underestimation</li> <li>Wet soil scattering</li>              |  |  |
| 4b    | >10 mm              | >3% | ${\sim}1~\mathrm{mm}$ | Snowpack ripening
Smooth surface                                |  <li>Uncertainty in spatiotemporal LWC/TWC</li> <li>Scattering from surface structures (melt-refreeze)</li> <li>Surface roughness underestimation</li>                                           |  |  |
| 5b    | >10 mm              | >3% | $\sim 3 \text{ mm}$   | Snowpack ripening
Increasing surface roughness                  |  <li>Uncertainty in spatiotemporal LWC/TWC</li> <li>Uncertainty in surface roughness measurements</li> <li>Uncertainty in IEM modeling</li>                                                      |  |  |
| 6b    | >10 mm              | >3% | ~1 mm                 | New snowfall on a wet snowpack
Well-developed surface roughness | - "Buried surface roughness"                                                                                                                                                                              |  |  |
| 7b    | >10 mm              | >3% | >4 mm                 | Wet snowpack
Fully-formed suncups                               |  <li>Uncertainty in spatiotemporal LWC/TWC</li> <li>Uncertainty in surface roughness measurements</li> <li>Uncertainty in IEM modeling</li>                                                      |  |  |

Reviewer Comment 1.8 — Table 4 – Similar comment to table 3.

Reply: We thank the reviewer for this suggestion. We improved the readability of all tables by adding gridlines.

Reviewer Comment 1.9 — Figure 11 – The bottom pane on this figure is unnecessary – it does not add anything to the interpretation. Simply providing the LWC for the top layer would be sufficient. Also, the LWC legend is unnecessary on the bottom. The RMSH legend is good.

**Reply**: We thank the reviewer for this comment. We made substantial changes to ex-Fig. 11 to address the points raised by Reviewer 2 (see **Fig. 8**), and we modified the figure according to these suggestions.

**Reviewer 2**

Reviewer Comment 2.1 — The radiometric accuracy for the backscatter data of the reference cell needs to be specified in detail. The reference to Marin et al. (2020) does not provide specific information on radiometric uncertainty for this particular cell of 20 m x 20 m extent. In S1 IW mode data it covers only about 4 independent samples (looks), resulting in high speckle-related uncertainty. Marin et al. apply for speckle reduction a multitemporal filter of 11 x 11 pixels and a gamma-MAP filter of 3 x 3 pixels. In both cases the radiometry is preserved if the window covers a homogeneous distributed target. This is not the case in the area surrounding of the refence cell. For the multitemporal filter different temporal response within the window may introduce additional radiometric biases for sigma-0 of individual pixels and dates.

**Reply:**

We thank the reviewer for this insightful comment. It allowed us to clarify important aspects of the radiometric uncertainty (particularly the impact of speckle) which should indeed be a part of this study.

In the revised version of the manuscript, L271-290, we added the following discussion.

(...) The nominal radiometric uncertainty of S1 falls in the range of  $3\sigma$ =1.0 dB, as indicated in several ESA validation campaigns [1-4]. However, the overall radiometric accuracy is also affected by a number of preprocessing steps, including (but not limited to) the application of despeckle filters, terrain correction and radiometric normalization (particularly challenging in mountain regions with complex topography), and thermal noise removal (important in conditions of high absorption, such as wet snow). In such conditions, a detailed specification becomes extremely complex and falls beyond the scopes of this paper. Nonetheless, since this study explores the multitemporal behavior of  $\sigma_0^{VV}$  over a target cell, it is relevant to mention speckle denoising. We used the filter proposed by [5] – a powerful yet relatively simple one to denoise multitemporal stacks, with a 11 imes11 pixels window. Similarly to local spatial multi-looking, its implementation involves local averages of intensity values for each date. Intuitively, this could potentially blur strong targets and edges, ultimately leading to a loss of resolution and impacting the overall multitemporal result. However, in conditions of dry snow, the snow cover and the position of the scatterers are stable, snow temperatures are well below 0°C and the soil should be mostly frozen, implying constrained variations in soil moisture. Under these conditions, the pixels we considered in our study exhibited an overall stable behavior. The same stability was observed during dry periods in summer. In these two cases, the standard deviation was within 1.0 dB, which aligns with the nominal radiometric uncertainty of S1. During the melting period, the primary source of radiometric uncertainty originates from the formation of LWC within the snowpack. As a consequence, the radar return signal from the same target cell changes over time, resulting in reduced temporal coherence and larger deviations in multitemporal statistics. As will be shown in the course of this study, LWC potentially exhibits high heterogeneity across a single resolution cell. Under such conditions, the estimation of radiometric uncertainty becomes particularly challenging. Without a precise reference for LWC, a rigorous uncertainty quantification is inherently difficult and lies beyond the scope of this work. (...)

On the other hand, **Fig. 6** illustrates the behavior for track 168 during the melting period, when differences in time are present due to wetting of the snow. As evident, the differences between various window sizes and the

unfiltered signals are minimal, justifying our pragmatic choice of an  $11 \times 11$  pixel window: this choice aims to reduce uncertainty due to speckle while (trying to) introducing minimal bias. Additionally, in the new version of the manuscript (L269), we clarified that, unlike in [6], the gamma-MAP filter was not applied in this study.

Reviewer Comment 2.2 — The representativeness of the S1 signatures of the reference cell in respect to the area in its surroundings should be assessed by analysing also data of other cells, preferably covering some larger area. Considering Fig. 2, it is unclear why cell 39 is used as single reference case. The data of cell 39 do not show a distinct incidence angle dependence and track 117 shows a different behaviour in the two years. The data of cell 18 of the two years, for example, are consistent. According to the elevation contour lines, this cell is located on an east-facing slope, and the two tracks of the ascending orbit show lower sigma-0 values, as expected for a back-slope. This may offers a possibility for studying impacts of the incidence angle.

**Reply**: We thank the reviewer for their comment regarding the representativeness of the selected S1 cell. These remarks helped us to corroborate the premises of our analyses.

In the manuscript (Sec. 4.2), we discussed the heterogeneity of LWC measurements (even when measuring simultaneously). Such heterogeneity has been previously analyzed and consolidated in both laboratory settings and field studies [7,8]. It is true that the representativeness of the chosen S1 reference cell would benefit from the comparisons to other neighboring cells. However, the range of locations within the field site being undisturbed, potentially matching the snow characteristics measured in the snow pits and not affected by double-bounce effects from big metallic structures nearby is rather small and likely limited to cells 18, 25, 32, 38, 39, 40. We added Fig. 2 to the manuscript, where the satellite signature from cell 40 is compared to that of this ensemble of similar cells.

In Section 3.1 of the manuscript, cell 39 was initially chosen primarily because it is the closest to our measurement field and therefore most likely to represent the snow conditions accurately. However, the points brought up by Reviewer 2 made us reconsider several points, which led us to change the reference cell to cell 40. We had three major concerns with the choice of cell 18 as a reference cell. In the first place, although it lies officially outside of the skiable domain, it is common for skiers to traverse that area, especially for employees on duty for daily snow measurements throughout the snow season. On the other hand, cell 18 belongs to a east-faced slope ( $\sim$ 11°), where the snow cover becomes patchy and disappears earlier with respect to the flatter area ( $\sim$ 5.5°) where the preferred cells are situated (see Fig. 7). This could introduce ambiguities in our discussion of backscatter increases related to the development of surface roughness. Finally, cell 18 lies  $\sim$ 90 meters away from the area where we performed our snow pits. To our knowledge, the greatest horizontal distance over which LWC variability has been characterized is 5 meters on relatively flat terrain [7]. Therefore, selecting a greater horizontal distance and slope difference than what we used could potentially introduce additional uncertainties in our analysis that we would be unable to quantify. In the revised version of the manuscript, L350-368, this choice is justified as follows.

(...) Ideally, the target cell should coincide with the location of in-situ measurements to ensure that the observed snow properties accurately represent those detected by the radar. Although S1 footprint is large relatively to the area disturbed by a single snow pit, excavating multiple consecutive snow profiles across a broader area can ultimately alter snow conditions across the entire cell – particularly under moist or wet snow conditions. This would introduce an uncontrolled degree of uncertainty. Therefore, the target cell should rather be selected among the surrounding undisturbed cells with similar slopes and aspect. Fig. 4a-b (Fig. 2a-b) show the dry-snow  $\sigma_0^{VV}$  variability for a set of cells with such features, i.e., cells 18, 25, 38, 39, 40. Among these, cell 40 shows a distinguished dependence on each incidence angle and orbit direction, along with relatively low variability of  $\sigma_0^{VV}$  across tracks. An exception occurs for track #117 during 2023-2024, where the variability is relatively higher with respect to the year before. This increased variability is also noticeable for cell 25 and 39. Given the lower variabilities recorded on the prior year, interference from non-natural elements can be ruled out. The most plausible explanation is a certain degree of heterogeneity in soil moisture across the field. Unfortunately, we are unable to verify this hypothesis, as soil moisture measurements were not included in our field campaign. Additionally, cell 40 lies in the immediate vicinity to the measurement site, and the snow surface remains undisturbed due to the operation of a LiDAR laser scanner continuously monitoring the snow surface. Fig. 4c-f (Fig. 2c-f) illustrates the

multitemporal  $\sigma_0^{VV}$  signal from cell 40 in comparison to that of the other candidate cells. The average standard deviation of the  $\sigma_0^{VV}$  ensemble across these cells is approximately 3 dB for all tracks. Interestingly, the lowest standard deviation is consistently observed at the time of the backscatter drop caused by wet snow, with the exception of track #117 in 2024. Notably, during the melting season, the signal from cell 40 lies in the lower end of the backscatter range across all years and tracks – aside for track #117 in 2022-2023. Potentially, this behavior is desirable for wet snow detectability. For these reasons, the  $\sigma_0^{VV}$  recorded over cell 40 is selected as the reference time series for this work. (...)

**With respect to the incidence angle (L369-375):**

(...) The impact of incidence angle was not a primary focus of this study, as it has already been extensively addressed in previous research [9–12], which strongly relied on tower-based instruments allowing greater control than satellite-based radar systems. In our case, the area most representative of measured snow properties is relatively small and flat, resulting in a limited range of local incidence angles available for analysis (see Fig. 2 (Fig. 9)). Furthermore, the high spatial variability of LWC would require dedicated reference measurements for each incidence angle and cell, which was not feasible given the time and resources already involved in conducting the campaign at a single location. (...)

Reviewer Comment 2.3 — For relating the backscatter modelling results to specific melt phases and for interpretation of the observed backscatter signatures, first of all it is important providing information on the backscatter contributions of the individual snow layers in dependence of the liquid water content. The relative contributions to total backscatter in dependence of depth below the snow surface should be specified for typical snow profiles shown in Figs. 5 and 6. Another concern is the limited information regarding properties of the scattering elements in the individual layers and the parametrization of the dense medium effect. In particular for the early melt phases, the properties of the top snow layer can be quite different, depending on size and shape of water inclusions (Arslan et al., 2003). Furthermore, information on the impact of incidence angle dependence of backscatter would be of interest, being of relevance for assessing roughness effects of wet snow surfaces.

**Reply**: We thank the reviewer for these valuable comments, which highlight issues that were not sufficiently addressed in the initial version of the manuscript.

In an earlier version of the manuscript, we included a section addressing the relative contribution to the total backscatter of individual snow layers, as a function of the measured liquid water content. We finally decided to remove it, because the solidity of such analysis is strongly dependent on two main aspects. One of them is the lack of a consolidated wet snow permittivity formulation, which as explained in Section 3.2 of the manuscript, remains an unresolved issue in the field of electromagnetic modeling of wet snow. In our work, we compare two formulations which were previously validated against real-world C-band data [13–15]. However, these two formulations diverge significantly at C-band, especially for what concerns the prediction of the imaginary part of the permittivity, which governs absorption losses. Such divergence becomes more pronounced for increasing values of LWC, as we show in Fig. 5 of the manuscript. On the other hand, we were unable to measure and model larger (superficial or internal) snow structures potentially causing additional scattering effects, such as crusts, which might have an important effect on the total recorded backscatter at the cell-scale. In the reviewed version of the manuscript, we highlighted this point in the Discussion, L804-817.

(...) Moreover, Fig. 12 (**Fig. 5**) highlights discrepancies of approximately 6 dB between SMRT-simulated and satellite-recorded backscatter signals, especially when  $\sigma_0^{VV}$  is largely dominated by LWC. Similar deviations were found by [16] using MEMLS&a to reproduce  $\sigma_0^{VV}$  during consecutive melt seasons over alpine areas. Additionally, both permittivity models saturate  $\sigma_0^{VV}$  at values below -30 dB. Such low values are never recorded by S1, which saturates at around -22 dB. Similar signal saturation (between -20 and -25 dB) in the C-band in vertical copolarizations are confirmed by the tower-based radiometric studies of [11, 12]. Matching the recorded S1  $\sigma_0^{VV}$  would require an imaginary part of  $\epsilon_s$  similar to that at 1 GHz – this would imply unrealistic penetration depths for the C-band, contradicting field observations [17–20]. We conclude that one possible explanation to the observed deviations is the overestimated absorption loss in the existing permittivity formulations. In view of the described inherent limitations of existing wet snow permittivity formulations, a detailed quantitative analysis of scattering contributions from individual snow layers was not possible. As previously noted in Sec. 3.2, the absence of a

unified permittivity model for wet snow remains an important direction for future research – not only for RT modelling, but also for field measurements, since dielectric methods depend on such models to derive LWC. (...)

The reviewer is right in pointing out that the original version of the manuscript lacked description of the parametrization of dense medium effects. In the revised version, we addressed this by adding a dedicated paragraph in Sec. 3.2 (L433-444).

(...) RT modelling of snow comes with the additional difficulty of quantifying the dense medium effects, i.e., the electromagnetic interactions occurring between snow grains that are closely packed together. At C-band frequencies, these effects become significant as the scattering regime changes due to the presence of liquid water – both through changes in snow grain interactions and in bulk dielectric properties. In H-86, dense medium effects are not accounted for. In MEMLSv3, these effects are accounted through a semi-empirical parametrization involving, among other parameters, correlation length, density-dependent corrections and – as mentioned above – mixing formulas. Correlation lengths are used to represent the effective grain size and spatial correlation of the ice matrix, and to capture the degree of interaction between dense grains. Despite the range of correlation lengths being limited in MEMLSv3, the ones that are represented derive from structures observed at Weissfluhjoch during two snow seasons [21]. Therefore, they are likely suitable to describe the dense medium effects on the snowpack structures observed and measured in this study. Snow density is used as a proxy to determine how closely grains are packed; and as density increases, scattering is reduced and absorption increases. Such corrections are embedded into the extinction term, i.e., the sum of scattering and absorption coefficients. (...)

We added further clarifications on the mixing formulas used in the two formulations in Sec. 3.2 of the revised manuscript (L410-414).

(...) For this study, we selected two formulations: (i) the Microwave Emission Model for Layered Snowpacks [21] in its 3rd version (MEMLSv3 hereafter), which is based on the Maxwell-Garnett mixing theory of dry snow and prolate water inclusions; (ii) the Debye-like model modified by [13] (H-86 hereafter), which uses a mixing formula based on volume fractions and refractive indices, calibrated against field data. These models were selected because they were validated against real-world C-band data. (...)

Furthermore, we agree with the reviewer that our dataset lacks information about the shape of the water inclusions per layer, and we thank them for providing the reference to [22], which we overlooked in our literature review. To our knowledge, neutron radiography can be used to distinguish water from ice, however, it requires either prior knowledge of the dry density or other measurements such as energy-selective neutron radiography [23]. In such experiments, the initial conditions need to be carefully controlled, therefore requiring a laboratory setting. [24] recently presented the nuclear MRI rapid profiling technique, which allowed measuring different states of wetting snow and therefore the shape of the water inclusions at unprecedented resolutions. This technique also depends on a controlled laboratory environment. At the time our measurement campaign was designed and conducted, these methods did not yet exist – let alone their applicability in the field, which is still entirely unknown. We addressed this in a new paragraph in the Discussion (L805-818).

(...) Finally, the lack of a definitive permittivity formulation for wet snow poses a significant challenge for the scientific community. The permittivity formulations selected for this study exhibit similar spectral shapes (see Fig. 5) and are, to our knowledge, the only ones that have been validated against real-world observations at C-band frequencies. As mentioned in Sec. 3.2, the permittivity formulation describes how the real and imaginary part of  $\epsilon_s$  change with increasing fractions of liquid water, and therefore how radar microwaves interact with the snowpack.  $\epsilon_s$  is computed using mixing theories to account for volume fractions of ice, water and air in the snow medium. MEMLSv3 parametrizes the shape of water inclusions as elongated spheroids embedded in a homogeneous host medium. This represents an important source of uncertainty. As liquid water increases, the shape and orientation of water inclusions significantly affects  $\epsilon_s$ , as the electromagnetic field interacts with them in a shape-dependent way, generating anisotropic responses [22, 25]. However, characterizing the temporal evolution of the shape of water inclusions during melting processes is an ambitious and challenging task that has only been addressed very recently by [24] through rapid MRI profiling in a controlled laboratory environment. At the time our measurement campaign was designed and conducted, these methods did not yet exist – let alone

their applicability in the field, which is still entirely unknown. These recent advancements are highly promising for the crucial challenge of developing a comprehensive model applicable across all frequencies and LWC conditions. (...)

Finally, the Reviewer correctly points out that in the original version of the manuscript, we didn't analyze incidence angle dependencies on roughness effects on wet snow surfaces. To this end, in the new version of the manuscript, we modified ex-Fig.11 with Fig. 8, also accounting comments on readability made by Reviewer 1. We added complementary paragraphs in the Results section 4.3.1 (L727-738) and in the Discussion (L879-899).

(...) Finally, Fig. 14c-f (**Fig. 8c,f**) allow considerations regarding the impact of the incidence angle. To do so, we use the index  $|\Delta\sigma_0^{VV}|_{40-30}$ , i.e., the absolute difference in backscatter between the two incidence angles of 40° and 30° – the range of incidence angles overlooking the reference cell. For smooth surfaces ( $1 \le RMSH \le 2$ ) and for LWC $\ge 1.5\%$ ,  $|\Delta\sigma_0^{VV}|$  exceeds 2 dB, i.e., twice the nominal uncertainty of S1 (see Sec. 2.3). For LWC lower than 1.5%,  $|\Delta\sigma_0^{VV}|$  is highly sensitive to small increases in LWC. For RMSH $\ge 3$ , the sensitivity of  $|\Delta\sigma_0^{VV}|$  to changes in LWC almost disappears. In conditions of fully-formed suncups (RMSH $\ge 10$ ),  $|\Delta\sigma_0^{VV}|$  drops below the nominal sensitivity of 1.0 dB for every LWC value, meaning that the backscatter signals show progressively weaker angular dependence for highly structured snow surfaces. This phenomenon is easily understood considering that, on rough surfaces, diffuse scattering is enhanced. Therefore, the position of the sensor relatively to the snow surface becomes less important, as the reflected energy is less directional and more broadly scattered. The same phenomenon explains the apparent slight backscatter decrease for RMSH $\ge 10$  at angles of 30° (Fig. 14b,e (Fig. 8b,e)). At lower incidence angles, the radar beam is closer to perpendicular to the surface than it is at higher incidence angles. On rough surfaces, with enhanced diffuse scattering, the fraction of energy reflected directly back to the sensor is reduced. (...)

[...]

(...) Fig. 14c,f (Fig. 8c,f) show that for smooth surfaces and for LWC values as low as 1.5% – i.e., when the melting process is likely in its initial stage - the variation in backscatter across the range of incidence angles overlooking the reference cell is comparable to or even exceeds the threshold used in [6, 26, 27] for wet snow detection. This angular dependence constitutes an additional uncertainty factor in wet snow detection, which overlaps with the previously discussed effects of diurnal variability in snowpack properties. On the other hand, for LWC values higher than 2% on smooth surfaces, the angular dependence increases up to 3 dB. This result supports the hypothesis that two distinct scattering mechanisms observed across the two seasons are directly linked to incidence angle effects. The first is a persistent 3–5 dB difference in  $\sigma_0^{VV}$  between the two ascending tracks, recorded from mid-April to early June 2023 (see Fig. 8a). This spread was not observed in the following year. Our LWC measurements indicate that the snowpack surface was wetter in 2023 than in 2024, likely due to the presence of ice lenses acting as drainage barriers for meltwater and favoring the formation of a wetter layer above them (see Fig. 13 and Tab. 5). Consequently, and in line with the results in Fig. 14 (Fig. 8), the smoother and wetter snow surface in 2023 led to a stronger angular dependence compared to 2024. Additionally, the angular dependence decreases with increasing surface roughness. The second observed feature is the sharp decrease in backscatter between consecutive acquisitions of both ascending and descending tracks in 2024 — from June 15 to 22 and from June 19 to 26, respectively. Our measurements indicate conditions of high snowpack saturation and surface roughness values equal to or exceeding 10 mm (see Fig. 13 and Tab. 5). Consistent with the results shown in Fig. 14b,e (Fig. 8b,e), we interpret this decrease as the result of suncups formation on a saturated snow surface. The enhanced surface roughness likely increased diffuse scattering and reduced the proportion of energy reflected back to the sensor, thereby explaining the observed backscatter decrease. These findings indicate that, despite all the aforementioned challenges in deriving LWC from backscatter and vice versa, the multitemporal analysis of angular dependence may carry valuable additional information. Unfortunately, further analysis in this direction was limited by the reduced revisit frequency of S1 during the period of this study. (...)

Reviewer Comment 2.4 — Introduction and Discussion: During three winters Strozzi and Mätzler (1997; 1998) performed at the same test field above Davos C- and Ka-band backscatter measurements. Reference 1 (1997) is briefly cited in the manuscript, reference 2 (1998) is not cited. Results of these measurements

are relevant within the context of the work presented by Carletti et al. and key points should be mentioned. Among issues addressed in the two papers are impacts of surface roughness and refrozen snow crusts based on measurements at different incidence angles, and the response to liquid water content.

**Reply**: We thank the reviewer for this comment – we have missed the above mentioned article in our literature review. In the revised version of the manuscript, we have strengthened the Introduction (L45-56) by referencing to the findings of this study.

(...) This raised the question of whether different types of snow cover could be classified based on their response to active microwave signals. This challenge has been addressed with various approaches over the years. Between 1993 and 1995, at the field site of Weissfluhjoch in the Swiss Alps, [11, 12] conducted tower-based C-band radiometric measurements at all polarizations across a wide range of incidence angles. Simultaneously, they carried out monthly measurements of snow physical properties. These measurements were used to classify the observed snow covers into categories ranging from dry snowpacks, to thin moist layers overlying dry snow, to wet snowpacks with either smooth or rough surfaces. Relying on a tower-based radiometer, the experiments were highly controlled, allowing detailed investigation of radar responses to each snow condition. Nevertheless, significant sources of uncertainty remained – especially the influence of surface roughness on wet snow surfaces, which was not quantitatively measured, but only qualitatively assessed. These detailed studies, along with the work of [15], raised questions about theoretical foundations and systematic reliability of LWC retrieval algorithms based on C-band full-polarimetric SAR imagery, which had been developed shortly before [18, 28]. In particular, the scattering behavior used for such retrieval algorithms could have been biased by a combination of conditions leading to a strong prevalence of surface scattering mechanisms. (...)

Moreover, in the Discussion section, we now refer to the results of these studies at several points (L790, L826, L852, and L875).

Reviewer Comment 2.5 — P4, L121: The sensitivity to snow wetness depends on the local incidence angle, not directly on slope steepness. On steep fore-slopes the sensitivity is low.

**Reply**: Thank you for this comment. We corrected the sentence to: (...) On the one hand,  $\sigma_0^{VV}$  is less sensitive to changes in snow wetness at low incidence angles [27] (...)

Reviewer Comment 2.6 — P8, L212: A cutter of 55 cm length may not be suitable for resolving the density differences between individual layers that may be quite thin during the different melt phases. Please explain the limits in vertical resolution.

**Reply**: The cylinder cutter was used for manual measurements of bulk snow water equivalent, not density. They served as validation for the measurements from the automatic snow scale. As explained in Section 2.1.2 of the manuscript, manual density measurements were performed using a box cutter with a vertical dimension of 3 cm. This method allowed us to precisely measure density differences between individual snow layers. An example of a full measured density profile is shown in Fig. 3a, and its corresponding discretization into physically consistent layers for SMRT is shown in Fig. 3b.

Reviewer Comment 2.7 — P9, Table 1: The elevation contour lines of Fig. 2a indicate different slope steepness within the test field. Please explain to which points the cited incidence angles refer and show the overall range of angles for the test field.

**Reply**: We thank the reviewer for this comment. In the original version, there was indeed ambiguity between "test field" and the reference cell actually chosen for the analysis. Following these suggestions, in the new version of the manuscript, we first show the overall range of incidence angles for the test field (see Fig. 9), and then we clarify that the range of incidence angles reported in Tab. 1 refers to the chosen reference cell, i.e., cell 40.

Reviewer Comment 2.8 — P11, Fig: 2: Incidence angles of cells on sloping show lower sigma-0 for ascending orbits, as to be expected for back-slopes. Consequently, the difference in sigma-0 between individual tracks may offer the possibility for exploring incidence angle effects.

**Reply**: We thank the reviewer for this comment. In Reply 2.2, we explained why studying the effects of the incidence angle is quite impractical in a study designed like ours, primarily due to the extreme heterogeneity of the LWC. However, we explored the effect of incidence angle for the range of angles overlooking the selected reference cell 40 – see Fig. 8 and Reply 2.3.

Reviewer Comment 2.9 — P12, L328: The characterization of snow microstructure is a critical issue for snow backscatter modelling. Exponential correlation functions have major deficiencies, in particular for multi-size and sticky cases (Chang et al., 2016). Furthermore, the phase functions of snow with liquid water inclusions are quite different from that of dry snow (Arslan et al., 2003).

**Reply**: We thank the reviewer for their comments about the characterization of snow microstructure, and for providing important literature reference which we overlooked.

The choice of an exponential auto-correlation function is based on the study of [29], specifically to Section 2.2, where the authors illustrate the unifying role of the microwave grain size ( $\ell_{MW}$ ) at low frequencies such as the C-band. In the revised version of the manuscript (L390-403), we improved the justification of such choice providing more details and addressing the deficiencies of exponential auto-correlation functions pointed out by the reviewer.

(...) Measurements of density and SSA were used to compute the Porod length  $(\ell_P)$  [30]. The microwave grain size  $(\ell_{MW})$  is computed as the product of  $\ell_P$  and the polydispersity k, a parameter describing the variability of the length scales with respect to the microstructure [29]. k was set to 0.75: this empirical value was estimated from  $\mu$ -CT scans of a wide variety of alpine snow samples with convex grains, among which rounded grains and melt forms [29]. For this study, snow microstructure was parametrized using the exponential model. For frequencies in the X- and Ku-bands (10-17 GHz), exponential auto-correlation functions have been shown to be too simplistic for representing snow microstructure. Their fast decay fails to capture long-range spatial correlations, and their inadequacy in modeling densely clustered media results in an underestimation of forward scattering effects [25]. However, [29] show how  $\ell_{MW}$  can be computed analytically for various forms of auto-correlation functions, including the exponential. These analytical expressions of  $\ell_{MW}$  allow for direct comparison between different representations of snow microstructure. Most importantly, when the same value of  $\ell_{MW}$  is used as input, all microstructure models give the same scattering amplitude in the low-frequency limit. Therefore, according to these findings, the choice of the best representation of snow microstructure becomes a secondary problem with respect to measuring  $\ell_{MW}$  in order to predict snow scattering in the C-band. (...)

Reviewer Comment 2.10 — P14, L363: In particular during a main part of winter 2023-2024 the base of the snowpack shows zero deg. temperature, implying unfrozen ground. This goes on throughout the snowmelt periods.

**Reply**: We thank the reviewer for this remark. The sentence they refer to was formulated poorly. Indeed, we did not model the soil as a frozen surface, but rather as a reflecting surface with a given backscatter value. In the revised version of the manuscript (L450-459), we clarified this point.

(...) Using the functions available in SMRT, we modeled the substrate as a reflecting surface with a given value of backscatter. In dry snow conditions, on days when manual measurements and satellite overlooks coincided, we assigned the S1 recorded backscatter value to the substrate, assuming that dry snow is transparent to radar waves at C-band and that therefore the soil is the only contribution to the total backscatter. In wet snow conditions (or in dry snow conditions, when there was no concomitance between measurements and satellite overlooks), we assigned a fixed value of backscatter to the substrate, which we computed as the average value in dry snow conditions of each individual track (incidence angle). Notably, SMRT offers the possibility to compute the backscatter from the soil, however, it requires a series of detailed information that are spatially heterogeneous and would have been nearly impossible to retrieve continuously over the course of our campaign. These properties include the soil moisture, the relative sand content, the relative clay content, the soil content in dry matter, and other geometrical parameters such as the roughness and the correlation length. (...)

Table 2: Overview on the identification of the melting phases based on the multitemporal S1 SAR backscatter as proposed by [6]. For each season, the table shows the relevant values of  $\sigma_0^{VV}$  and the occurrence dates for each afternoon/ascending (A↑) and morning/descending (M↓) look (and corresponding incidence angle). The selected values for the start of the moistening, ripening and runoff phases are highlighted in bold. For the runoff start, the selected date according to the method of [6] is compared against the data recorded by the lysimeter, when available.

| Season                                | 2022-2023             |          |          |          | 2023-2024              |          |          |          |
|---------------------------------------|-----------------------|----------|----------|----------|------------------------|----------|----------|----------|
| Track                                 | 015 (A↑)              | 117 (A↑) | 066 (M↓) | 168 (M↓) | 015 (A↑)               | 117 (A↑) | 066 (M↓) | 168 (M↓) |
| Local Incidence Angle                 | 41                    | 32       | 33       | 42       | 41                     | 32       | 33       | 42       |
| $\overline{\sigma^{VV}_{0,dry}}$ [dB] | -12.3                 | -11.4    | -8.4     | -10.0    | -12.6                  | -11.5    | -8.9     | -10.1    |
| Moistening start date                 | Apr 22                | Apr 29   | _        | _        | Apr 04 – Apr 16 | Mar 18   | _        | _        |
| Moistening start value [dB]           | -18.5                 | -16.3    | _        | _        | -14.120.0              | -13.9    | _        | _        |
| Ripening start date                   | _                     | _        | Apr 26   | Mar 28   | -                      | _        | Apr 08   | Apr 15   |
| Ripening start value [dB]             | _                     |          | -12.6    | -13.3    | _                      | _        | -12.8    | -17.9    |
| $\sigma_{0,min}^{VV}$ , date          | May 16                | Apr 29   | May 08   | May 03   | May 22                 | May 17   | May 26   | Jun 02   |
| $\sigma_{0,min}^{VV}$ , value [dB]    | -21.4                 | -16.3    | -19.8    | -22.4    | -22.6                  | -23.7    | -20.7    | -22.8    |
| Runoff start date [6]                 | May 06                |          |          |          | May 24                 |          |          |          |
| Runoff start date (Lysimeter)         | No data – ~Apr 29 (?) |          |          |          | ∼Apr 15                |          |          |          |

Reviewer Comment 2.11 — P16 Fig. 5: Between mid-April and early June 2023 sigma-0 of the two ascending tracks shows consistent differences of 3 to 5 dB. Please explain the reason. The high sigma-0 values are probably from track 117 which shows high variance in 2023-2024, differing from 2022-2023 (Fig. 2b).

**Reply**: We thank the reviewer for this remark. We explained this instance as an angular dependence in the course of the Discussion (L883-891).

(...) On the other hand, for LWC values higher than 2% on smooth surfaces, the angular dependence increases up to 3 dB. This result supports the hypothesis that two distinct scattering mechanisms observed across the two seasons are directly linked to incidence angle effects. The first is a persistent 3–5 dB difference in  $\sigma_0^{VV}$  between the two ascending tracks, recorded from mid-April to early June 2023 (see Fig. 8a). This spread was not observed in the following year. Our LWC measurements indicate that the snowpack surface was wetter in 2023 than in 2024, likely due to the presence of ice lenses acting as drainage barriers for meltwater and favoring the formation of a wetter layer above them (see Fig. 13 and Tab. 5). Consequently, and in line with the results in Fig. 14 (Fig. 8), the smoother and wetter snow surface in 2023 led to a stronger angular dependence compared to 2024. Additionally, the angular dependence decreases with increasing surface roughness. (...)

Reviewer Comment 2.12 — P20, Table 2: Please specify the incidence angle. Besides, the validity of these numbers in respect to other incidence angles would be of interest.

**Reply**: We thank the reviewer for this comment. In the original manuscript, the values in Tab. 2 were averaged over the 4 incidence angles. In the revised version, we improved this analysis analyzing all incidence angles (see Tab. 2).

The values in Tab. 2 are discussed in a new paragraph (L557-565).

(...) Because for the selected cell two morning/descending and afternoon/ascending looks are available, there are two possible dates for the start of the moistening and ripening phase, respectively. In 2023, these dates are Apr 22 and 29 for the moistening phase and Mar 28 and Apr 26 for the ripening phase. For the start of the moistening phase, we selected the earliest, i.e. Apr 22. For the start of the ripening phase, the two identified dates are almost one month apart, however, the  $\sigma_0^{VV}$  decrease recorded on Mar 28 by track #168 derives from a melt-refreeze cycle, as the following value recorded by the same track aligns back around the winter mean. Therefore, we selected Apr 26 as the start of the ripening phase. In 2024, for the moistening phase, the  $\sigma_0^{VV}$  value recorded on Apr 04 by track #015 is only 1.5 dB lower than  $\overline{\sigma_{0,dry}^{VV}}$ , however, the next passage of the same track on Apr 16 recorded a drop of already 7.4 dB. Therefore, the moistening start has been placed on Apr 04.

On this date, track #117 recorded a drop of 7 dB with respect to  $\overline{\sigma_{0,dry}^{VV}}$ . For the ripening start, we chose Apr 15. (...)

Reviewer Comment 2.13 — P 27, Fig. 11: Please specify the incidence angle. Strozzi and Mätzler (1997) show the incidence angle dependence of backscatter of wet snow (for smooth and rough surfaces) based on backscatter measurements at the same test site.

Reply: We thank the reviewer for this comment. This point was already addressed it in Reply 2.3.

Reviewer Comment 2.14 — P29, L625: The development of surface roughness after start of the snowmelt period depends also on the sequence and intensity of snowfall events, varying from year to year.

Reply: We thank the reviewer for this remark. We expanded the sentence accordingly (L748-752).

Reviewer Comment 2.15 — P29, L641ff: The parametrization of the scattering elements may as well be a reason for differences between recorded and modelled backscatter (see e.g. Arslan et al., 2003; Chang et al., 2016).

Reply: We thank the reviewer for this comment. This point was already addressed it in Reply 2.3.

**References**

- [1] R. Torres, P. Snoeij, D. Geudtner, D. Bibby, M. Davidson, E. Attema, P. Potin, B. Rommen, N. Floury, M. Brown, I. N. Traver, P. Deghaye, B. Duesmann, B. Rosich, N. Miranda, C. Bruno, M. L'Abbate, R. Croci, A. Pietropaolo, M. Huchler, and F. Rostan, "Gmes sentinel-1 mission," Remote Sensing of Environment, vol. 120, pp. 9–24, 2012, the Sentinel Missions New Opportunities for Science. [Online]. Available: https://www.sciencedirect.com/science/article/pii/S0034425712000600
- [2] N. Miranda, P. Meadows, G. Hajduch, A. Pilgrim, R. Piantanida, D. Giudici, D. Small, A. Schubert, R. Husson, P. Vincent, A. Mouche, H. Johnsen, and G. Palumbo, "The sentinel-1a instrument and operational product performance status," in 2015 IEEE International Geoscience and Remote Sensing Symposium (IGARSS), 2015, pp. 2824–2827.
- [3] M. Schwerdt, K. Schmidt, N. Tous Ramon, P. Klenk, N. Yague-Martinez, P. Prats-Iraola, M. Zink, and D. Geudtner, "Independent system calibration of sentinel-1b," *Remote Sensing*, vol. 9, no. 6, 2017.
- [4] H.-J. F. Benninga, R. van der Velde, and Z. Su, "Sentinel-1 soil moisture content and its uncertainty over sparsely vegetated fields," *Journal of Hydrology X*, vol. 9, p. 100066, 2020. [Online]. Available: https://www.sciencedirect.com/science/article/pii/S2589915520300171
- [5] S. Quegan and J. J. Yu, "Filtering of multichannel sar images," IEEE Transactions on Geoscience and Remote Sensing, vol. 39, no. 11, pp. 2373–2379, 2001.
- [6] C. Marin, G. Bertoldi, V. Premier, M. Callegari, C. Brida, K. Hürkamp, J. Tschiersch, M. Zebisch, and C. Notarnicola, "Use of sentinel-1 radar observations to evaluate snowmelt dynamics in alpine regions," *The Cryosphere*, vol. 14, no. 3, pp. 935–956, 2020. [Online]. Available: https://tc.copernicus.org/articles/14/935/2020/
- [7] F. Techel and C. Pielmeier, "Point observations of liquid water content in wet snow ndash; investigating methodical, spatial and temporal aspects," *The Cryosphere*, vol. 5, no. 2, pp. 405–418, 2011. [Online]. Available: https://tc.copernicus.org/articles/5/405/2011/

- [8] F. Avanzi, H. Hirashima, S. Yamaguchi, T. Katsushima, and C. De Michele, "Observations of capillary barriers and preferential flow in layered snow during cold laboratory experiments," *The Cryosphere*, vol. 10, no. 5, pp. 2013–2026, 2016. [Online]. Available: https://tc.copernicus.org/articles/10/2013/ 2016/
- [9] C. Mätzler, "Applications of the interaction of microwaves with the natural snow cover," Remote Sensing Reviews, vol. 2, no. 2, pp. 259–387, 1987. [Online]. Available: https://doi.org/10.1080/02757258709532086
- [10] J. Shi and J. Dozier, "Radar backscattering response to wet snow," in [Proceedings] IGARSS '92 International Geoscience and Remote Sensing Symposium, vol. 2, 1992, pp. 927–929.
- [11] T. Strozzi, A. Wiesmann, and C. Mätzler, "Active microwave signatures of snow covers at 5.3 and 35 ghz," *Radio Science*, vol. 32, no. 2, pp. 479–495, 1997.
- [12] T. Strozzi and C. Matzler, "Backscattering measurements of alpine snowcovers at 5.3 and 35 ghz," *IEEE Transactions on Geoscience and Remote Sensing*, vol. 36, no. 3, pp. 838–848, 1998.
- [13] M. Hallikainen, F. Ulaby, and M. Abdelrazik, "Dielectric properties of snow in the 3 to 37 ghz range," *IEEE Transactions on Antennas and Propagation*, vol. 34, no. 11, pp. 1329–1340, 1986.
- [14] T. Achammer and A. Denoth, "Snow dielectric properties: from dc to microwave x-band," *Annals of Glaciology*, vol. 19, p. 92–96, 1994.
- [15] J. Kendra, K. Sarabandi, and F. Ulaby, "Radar measurements of snow: Experiment and analysis," Geoscience and Remote Sensing, IEEE Transactions on, vol. 36, pp. 864 879, 06 1998.
- [16] G. Veyssière, F. Karbou, S. Morin, M. Lafaysse, and V. Vionnet, "Evaluation of sub-kilometric numerical simulations of c-band radar backscatter over the french alps against sentinel-1 observations," *Remote Sensing*, vol. 11, no. 1, 2019. [Online]. Available: https://www.mdpi.com/2072-4292/11/1/8
- [17] F. T. Ulaby and W. Herschel Stiles, "Microwave response of snow," Advances in Space Research, vol. 1, no. 10, pp. 131–149, 1981. [Online]. Available: https://www.sciencedirect.com/science/article/pii/0273117781903896
- [18] J. Shi and J. Dozier, "Inferring snow wetness using c-band data from sir-c's polarimetric synthetic aperture radar," *IEEE Transactions on Geoscience and Remote Sensing*, vol. 33, no. 4, pp. 905–914, 1995.
- [19] F. Ulaby, D. Long, W. Blackwell, C. Elachi, A. Fung, C. Ruf, K. Sarabandi, J. Zyl, and H. Zebker, Microwave Radar and Radiometric Remote Sensing, 01 2014.
- [20] M. Lodigiani, C. Marin, and M. Pasian, "Mixed analytical-numerical modeling of radar backscattering for seasonal snowpacks," *IEEE Journal of Selected Topics in Applied Earth Observations and Remote Sensing*, vol. 18, pp. 3461–3471, 2025.
- [21] A. Wiesmann and C. Mätzler, "Microwave emission model of layered snowpacks," Remote Sensing of Environment, vol. 70, no. 3, pp. 307–316, 1999. [Online]. Available: https://www.sciencedirect.com/science/article/pii/S0034425799000462
- [22] A. Arslan, H. Wang, J. Pulliainen, and M. Hallikainen, "Scattering from wet snow by applying strong fluctuation theory," *Journal of Electromagnetic Waves and Applications*, vol. 17, no. 7, pp. 1009–1024, 2003.
- [23] M. Lombardo, A. Fees, A. Udke, K. Meusburger, A. van Herwijnen, J. Schweizer, and P. Lehmann, "Capillary suction across the soil-snow interface as a mechanism for the formation of wet basal layers under gliding snowpacks," *Journal of Glaciology*, p. 1–46, 2025.

- [24] Q. Krol, E. Scherrer, M. Skuntz, S. Codd, A. Hansen, and J. Seymour, "Rapid mri profiling of liquid water content in snow: Melt and stability during first wetting and rain on snow events," in *Proceedings of the International Snow Science Workshop*. Tromsø, Norway: Montana State University Library, 2024. [Online]. Available: https://arc.lib.montana.edu/snow-science/objects/ISSW2024\_O3.11.pdf
- [25] W. Chang, K.-H. Ding, L. Tsang, and X. Xu, "Microwave scattering and medium characterization for terrestrial snow with qca-mie and bicontinuous models: Comparison studies," *IEEE Transactions on Geoscience and Remote Sensing*, vol. 54, no. 6, pp. 3637–3648, 2016.
- [26] T. Nagler and H. Rott, "Retrieval of wet snow by means of multitemporal sar data," *IEEE Transactions on Geoscience and Remote Sensing*, vol. 38, no. 2, pp. 754–765, 2000.
- [27] T. Nagler, H. Rott, E. Ripper, G. Bippus, and M. Hetzenecker, "Advancements for snowmelt monitoring by means of sentinel-1 sar," *Remote Sensing*, vol. 8, no. 4, 2016. [Online]. Available: https://www.mdpi.com/2072-4292/8/4/348
- [28] J. Shi, J. Dozier, and H. Rott, "Deriving snow liquid water content using c-band polarimetric sar," in *Proceedings of IGARSS '93 IEEE International Geoscience and Remote Sensing Symposium*, 1993, pp. 1038–1041 vol.3.
- [29] G. Picard, H. Löwe, F. Domine, L. Arnaud, F. Larue, V. Favier, E. Le Meur, E. Lefebvre, J. Savarino, and A. Royer, "The microwave snow grain size: A new concept to predict satellite observations over snow-covered regions," AGU Advances, vol. 3, no. 4, p. e2021AV000630, 2022, e2021AV000630 2021AV000630. [Online]. Available: https://agupubs.onlinelibrary.wiley.com/doi/abs/10.1029/2021AV000630
- [30] G. Porod, "Die röntgenkleinwinkelstreuung von dichtgepackten kolloiden systemen," Kolloid-Zeitschrift, vol. 124, no. 2, pp. 83–114, 1951. [Online]. Available: https://doi.org/10.1007/BF01512792

Figure 2: Variability of  $\sigma_0^{VV}$  in dry snow conditions for all relative orbits overlooking cells 18, 25, 32, 38, 39, 40, i.e. the flat terrain cells with likely similar snow properties as the measured ones (a-b). Multitemporal  $\sigma_0^{VV}$  signal of the selected cell 40 compared to the ensemble standard deviation  $(std_{\sigma_0^{VV}})$  of the similar cells – morning/descending (M $\downarrow$ ) and afternoon/ascending (A $\uparrow$ ) (c-f).

Figure 3: (a) Vertical profiles of snowpack properties measured in the field on May 14, 2024: temperature (dark red), density (dark yellow), liquid water content (LWC; light blue), and specific surface area (SSA; dark blue). The vertical spacing of the points connected by the lines reflects the measurement resolution for each profile: 5 cm for temperature, 3 cm for density, 2 cm for LWC, and 4 cm for SSA. (b) Representation of the same profiles averaged according to the physically consistent snow layers (indicated by grey horizontal lines). The layered profiles as in (b) form the input snowpack for the SMRT model, combined with surface roughness parameters measured on the same day (RMSH=2.7 mm; CL=48.5 mm).

Figure 4: Bias between LWC measurements with dielectric devices and melting calorimetry for snow seasons of 2023 (a) and 2024 (b). In 2024, direct comparisons between simultaneous (brown) and co-located (light blue) measurements were also performed.

Figure 5: Comparison between the recorded multitemporal S1  $\sigma_0^{VV}$  (triangles and shaded areas) and the time series of  $\sigma_0^{VV}$  modelled with SMRT, for year 2023 (a) and 2024 (b). Results are shown for both permittivity formulations – MEMLSv3 (dark gray boxplots) and H-86 (light gray boxplots). The boxplots indicate the variability associated to the LWC uncertainty of  $\pm 1\%$  for each layer, as discussed in Sec. 4.2. The shaded areas of the recorded S1 multitemporal  $\sigma_0^{VV}$  represent the range of values obtained by connecting the consecutive passages by direction of orbits, i.e. by connecting all the morning/descending and the afternoon/ascending acquisitions. The triangles represent the exact values of the acquisitions. For clarity, exact values are only shown for days where snow measurements were carried out, thus allowing direct comparison. Colored boxes group similar simulation results and are labeled with codes (e.g., 1a, 2a), which refer to Tab. 3 (Tab. 1) for details on the corresponding measured snow properties, dominant scattering mechanisms, and potential sources of error. At the top of each panel, the time series are further segmented into the melting phases identified in Sec. 4.1 – as well as the main scattering regimes, which are influenced by LWC, surface roughness, and buried surface roughness.

Figure 6: Effect of the multitemporal filter, with different window sizes, to the backscatter signal for track 168 during the melting season of 2023-2024.

Figure 7: Webcam acquisition of the field site of Weissfluhjoch (June 11th, 2025 at 16:00:00). Area 1 (in blue) indicates the approximate location of cell 18. During snow ablation, this section shows earlier snow disappearance with respect to the flatter locations where cells 32, 38, 39, 40 belong, highlighted by area 2 (in red).

Figure 8: Sensitivity of the C-band radar backscatter to the coupled evolution of surface roughness (expressed by RMSH) and LWC. Panels (a, b, d, e) illustrate differences between two dielectric permittivity formulations – MEMLSv3 (a, b) and H-86 (c, d) – as well as the sensitivity to the local incidence angle (LIA) over cell 40.  $\sigma_0^{VV}$  responses are shown for 40° (solid lines) and 30° (dotted lines) incidence angles. Panels (c, f) show values of  $\left|\Delta\sigma_0^{VV}\right|_{40-30}$ , i.e., the differences between backscatter coefficients in (a, b) and (d, e), respectively. The real reference case is the snowpack layering observed on Apr 16, 2024: a melt event in the superficial 45 cm and an otherwise dry snowpack. The reported results are consecutive synthetic variations of LWC and roughness of the surface layer.

Figure 9: Overall range of local incidence angles across the study area for all the four relative orbits – morning/descending ( $M\downarrow$ ) and afternoon/ascending ( $A\uparrow$ ). Each S1 cell is identified by its centroid and a number.